# Adaptive Sharpness-Aware Minimization with a Polyak-type Step size: A Theory-Grounded Scheduler

**Dimitris Oikonomou** [1] [2]  **Nicolas Loizou** [1] [3]

## Abstract

Sharpness-Aware Minimization (SAM) has established itself as a powerful and widely adopted optimizer for training machine learning models. By explicitly minimizing the sharpness of the loss landscape, SAM often improves generalization while delivering strong empirical performance. However, SAM and its variants, like most training algorithms, are sensitive to the choice of learning rate, which is typically selected through extensive hyperparameter tuning or predefined schedulers. In this work, motivated by recent advances on the effectiveness of stochastic Polyak step sizes for Stochastic Gradient Descent (SGD), we derive Polyak schedulers tailored to SAM-style updates, yielding novel adaptive algorithms in both deterministic and stochastic settings. In the smooth setting, we prove linear convergence for strongly convex objectives and an $\mathcal{O}(1/T)$ convergence rate for convex objectives in the deterministic case. In the stochastic setting, we establish analogous convergence guarantees up to a neighborhood of the optimum. Numerical experiments demonstrate that the proposed Polyak schedulers achieve performance comparable to or better than carefully tuned SAM baselines, while substantially reducing the need for learning-rate tuning.

## 1. Introduction

A central challenge in modern machine learning is to explain and predict the generalization behavior of Deep Neural Networks (DNNs) (Zhang et al., 2016; Hardt et al., 2016;

---
[1]Mathematical Institute for Data Science (MINDS), Johns Hopkins University, Baltimore, MD, USA [2]Department of Computer Science, Johns Hopkins University, Baltimore, MD, USA [3]Department of Applied Mathematics and Statistics, Johns Hopkins University, Baltimore, MD, USA. Correspondence to: Dimitris Oikonomou, Nicolas Loizou <doikono1@jh.edu, nloizou@jhu.edu>.

*Proceedings of the 43rd International Conference on Machine Learning*, Seoul, South Korea. PMLR 306, 2026. Copyright 2026 by the author(s).

Neyshabur et al., 2017; Wilson et al., 2017; Neyshabur et al., 2018; Zhang et al., 2021). In the deep learning regimes that arise in practice, the empirical risk landscape often contains many stationary points that fit the training data (Liu et al., 2020). While these solutions may achieve similarly low training loss, their performance on unseen data can differ substantially. This suggests that the choice of optimization algorithm can influence which solution is reached and, consequently, the model's generalization performance (Foret et al., 2021).

A useful way to view this phenomenon is through the local geometry of the loss landscape: empirical evidence indicates that the sharpness of the training loss, i.e., how much the loss changes under small perturbations of the parameters, often correlates with generalization performance (Keskar et al., 2016; Dziugaite & Roy, 2018; Jiang et al., 2019; Singh et al., 2025). This observation has motivated the development of methods that seek to control or reduce sharpness as a means to improve generalization (Wu et al., 2020; Foret et al., 2021; Zheng et al., 2021; Andriushchenko et al., 2023; Xie et al., 2024a;b; Tahmasebi et al., 2024). Sharpness-Aware Minimization (SAM) and related variants are prominent examples of such algorithms (Foret et al., 2021) which avoid sharp minima by evaluating the loss under a small perturbation of the weights in each iteration and then take a step that improves this perturbed objective. In many settings, SAM-style updates yield better generalization across architectures and benchmarks, but they remain highly sensitive to the choice of the learning rate.

In this work, we focus on alleviating this issue and designing SAM variants that do not require tuning their learning rate. In particular, we consider both deterministic and stochastic optimization problems (formulation of the training objective). In the deterministic setting, our goal is to minimize the objective function

$$\min_{x \in \mathbb{R}^d} f(x), \tag{1}$$

where $f : \mathbb{R}^d \to \mathbb{R}$ is differentiable, lower bounded, $L$-smooth, and convex (or $\mu$-strongly convex). In the finite-sum setting (empirical risk minimization), the function $f$

has the form

$$f(x) = \frac{1}{n} \sum_{i=1}^{n} f_i(x), \tag{2}$$

where each component function $f_i : \mathbb{R}^d \to \mathbb{R}$ is differentiable, lower bounded, convex, and $L_i$-smooth. Let $X^*$ denote the set of minimizers of $f$. Throughout this work, we assume that $X^* \neq \emptyset$ and let $x^* \in X^*$ with $f^* = f(x^*) = \min_{x \in \mathbb{R}^d} f(x)$. The formulation (2) is the cornerstone of many machine learning tasks (Hastie et al., 2009), where the vector $x$ represents the model parameters, $f_i(x)$ is the loss associated with the training point $i$, and the goal is to minimize the empirical risk $f(x)$ across all training points.

To solve the empirical risk minimization problem (2), Foret et al. (2021) introduced the empirical sharpness at $x$ as $\max_{\|\varepsilon\| \leq \rho} [f(x + \varepsilon) - f(x)]$ and reformulated the finite-sum objective in (2) as the min–max problem:

$$\min_{x \in \mathbb{R}^d} \max_{\|\varepsilon\| \leq \rho} f(x + \varepsilon),$$

where $\varepsilon$ denotes a perturbation constrained to lie in a neighborhood of radius $\rho$. We refer to $\rho$ as the *sharpness radius*. The advantage of this viewpoint is that it effectively penalizes the empirical sharpness, thereby encouraging convergence to flatter minima. Approximating the inner maximization via a first-order Taylor expansion of $f$ around $x$ and optimizing over $\varepsilon$ yields the (Normalized) Sharpness-Aware Minimization (SAM) update:

$$e^t = x^t + \rho_t \frac{\nabla f_{S_t}(x^t)}{\|\nabla f_{S_t}(x^t)\|},$$
$$x^{t+1} = x^t - \gamma_t \nabla f_{S_t}(e^t), \tag{SAM}$$

where $S_t \subseteq [n]$ is a randomly sampled mini-batch of fixed size $|S_t| = \tau$, drawn independently at each iteration $t$. We use the notation $f_{S_t}(x) = \frac{1}{\tau} \sum_{i \in S_t} f_i(x)$ and $\nabla f_{S_t}(x) = \frac{1}{\tau} \sum_{i \in S_t} \nabla f_i(x)$. The normalization in the inner step ensures that the perturbation has controlled magnitude, since $\|e^t - x^t\| = \rho_t$. This constraint keeps the perturbed point close to $x^t$ and is known to yield more stable optimization, see Dai et al. (2023) for further discussion.

Building on SAM, Unnormalized Sharpness-Aware Minimization (USAM), given by

$$e^t = x^t + \rho_t \nabla f_{S_t}(x^t),$$
$$x^{t+1} = x^t - \gamma_t \nabla f_{S_t}(e^t). \tag{USAM}$$

was proposed in Andriushchenko & Flammarion (2022) and further analyzed in Shin et al. (2025); Dai et al. (2023). Unlike SAM, the iterate perturbation in USAM is not normalized, and therefore $e^t$ may lie substantially farther from

$x^t$. In turn, the resulting updates can be significantly more aggressive, which may render USAM less stable in practice. Nevertheless, in the original USAM work, Andriushchenko & Flammarion (2022) argue that such normalization is not essential for obtaining generalization improvements, and instead focus on USAM because it has better theoretical guarantees. In our work, we follow this approach and develop a theory primarily for the USAM update rule.

In this work, for SAM and USAM, we use the same mini-batch $S_t$ to compute the extrapolated point $e^t$ and to form the update $x^{t+1}$. This choice is standard in the SAM literature, though some works consider alternative sampling schemes, see Andriushchenko & Flammarion (2022). Moreover, note that by setting $S_t = [n]$ the update rules recover the deterministic (full-batch) versions of SAM and USAM. Finally, when $\rho_t = 0$, both SAM and USAM reduce to standard SGD (Gower et al., 2019; 2021).

**On Convergence of SAM-type methods.** A substantial body of work has analyzed the convergence behavior of SAM and USAM across a range of settings. In the deterministic regime, Dai et al. (2023) show that, for smooth and strongly convex objectives, SAM converges at a linear rate to a neighborhood of the optimum. Also in the deterministic case, Khanh et al. (2024) derive several basic guarantees for both SAM and USAM, including properties such as stationarity of accumulation points and convergence of gradient norms to zero. Beyond the purely deterministic case, Si & Yun (2023) provide convergence results for (SAM) in both deterministic and stochastic settings, covering convex, strongly convex, and non-convex objectives. In the stochastic regime, Andriushchenko & Flammarion (2022) establish convergence results for USAM under PL objectives as well as for general non-convex problems. More recently, Oikonomou & Loizou (2025b) develop a broad framework by studying a Unified SAM formulation that encompasses both SAM and USAM; under a relaxed condition (Expected Residual), they prove convergence guarantees in the stochastic regime, including linear rates for PL objectives and sublinear rates in the general non-convex setting.

In practice, a major limitation of SAM and its variants is the substantial effort required to tune its hyperparameters, which can be both costly and time-consuming. This has led to increasing interest in adaptive SAM-type methods, which adjust their parameters on the fly using information gathered during the iterations. For instance, Naganuma et al. (2024) propose an adaptive step size rule for SAM, based on the adaptive step size strategy of Malitsky & Mishchenko (2020), but do not establish convergence guarantees. Sun et al. (2024) introduce an adaptive learning-rate scheme for SAM together with momentum-based acceleration, and prove an $\mathcal{O}\left(1/\sqrt{T}\right)$ convergence

*Table 1.* Summary of various convergence guarantees for SAM and its variants. All the rates are stated for smooth objectives. The term $N$ indicates convergence up to a neighborhood of the solution; its definition varies across works (see the corresponding papers for details).

| Work | Problem Class | Step Size $\gamma_t$ | Assumptions on Noise | Rate |
|---|---|---|---|---|
| **Deterministic Setting** | | | | |
| (Dai et al., 2023) | $\mu$-Strongly Convex | Constant | - | $f(x^T) - f^* \le \mathcal{O}\left((1 - \gamma(2 - L\rho)\mu)^T + N\right)$ |
| (Si & Yun, 2023) | $\mu$-Strongly Convex | Decreasing | - | $f(x^T) - f^* \le \mathcal{O}\left(\exp(-T) + 1/T^2\right)$ |
| | Convex | Decreasing | - | $\frac{1}{T}\sum_{t=0}^{T-1}\|\nabla f(x^t)\|^2 \le \mathcal{O}\left(1/T + 1/\sqrt{T}\right)$ |
| | Non-Convex | Constant | - | $\frac{1}{T}\sum_{t=0}^{T-1}\|\nabla f(x^t)\|^2 \le \mathcal{O}\left(1/T + N\right)$ |
| (Oikonomou & Loizou, 2025b) | $\mu$-PL | Constant | - | $f(\overline{x}^T) - f^* \le \mathcal{O}\left((1 - \gamma\mu)^T\right)$ |
| Ours | Convex | *Adaptive* | - | $f(\overline{x}^t) - f^* \le \mathcal{O}\left(1/T\right)$ |
| | $\mu$-Strongly Convex | *Adaptive* | - | $\|x^T - x^*\|^2 \le \mathcal{O}\left((1 - \mu(1 - L\rho)^2/(4L))^T\right)$ |
| **Stochastic Setting** | | | | |
| (Si & Yun, 2023) | $\mu$-Strongly Convex | Decreasing | Bounded Variance | $\mathbb{E}[f(x^T) - f^*] \le \mathcal{O}\left(\exp(-T) + 1/T + N\right)$ |
| | Convex | Decreasing | Bounded Variance | $\frac{1}{T}\sum_{t=0}^{T-1}\mathbb{E}\|\nabla f(x^t)\|^2 \le \mathcal{O}\left(1/T + 1/\sqrt{T} + N\right)$ |
| | Non-Convex | Decreasing | Bounded Variance | $\frac{1}{T}\sum_{t=0}^{T-1}\mathbb{E}\|\nabla f(x^t)\|^2 \le \mathcal{O}\left(1/T + 1/\sqrt{T} + N\right)$ |
| (Sun et al., 2024) | Non-convex | *Adaptive* | Bounded Variance, Bounded Gradients | $\frac{1}{T}\sum_{t=0}^{T-1}\mathbb{E}\|\nabla f(x^t)\|^2 \le \mathcal{O}\left(1/\sqrt{T}\right)$ |
| (Oikonomou & Loizou, 2025b) | Non-convex | Decreasing | Expected Residual | $\frac{1}{T}\sum_{t=0}^{T-1}\mathbb{E}\|\nabla f(x^t)\|^2 \le \mathcal{O}\left(1/\sqrt{T}\right)$ |
| | $\mu$-PL | Constant | Expected Residual | $\mathbb{E}[f(x^T) - f^*] \le \mathcal{O}\left((1 - \gamma\mu)^T + N\right)$ |
| | $\mu$-PL | Decreasing | Expected Residual | $\mathbb{E}[f(x^T) - f^*] \le \mathcal{O}\left(1/T\right)$ |
| (Cheng et al., 2025) | Non-convex | *Adaptive* | Growth Condition | $\frac{1}{T}\sum_{t=0}^{T-1}\mathbb{E}\|\nabla f(x^t)\| \le \mathcal{O}\left(\log T/T^{1/4}\right)$ |
| Ours | Convex | *Adaptive* | - | $\mathbb{E}[f(\overline{x}^t) - f^*] \le \mathcal{O}\left(1/T + N\right)$ |
| | $\mu$-Strongly Convex | *Adaptive* | - | $\mathbb{E}\|x^T - x^*\|^2 \le \mathcal{O}\left((1 - \mu(1 - L_{\max}\rho)^2/(4L_{\max}))^T + N\right)$ |

rate for $\frac{1}{T}\sum_{t=0}^{T-1}\mathbb{E}\|\nabla f(x^t)\|^2$ on smooth objectives under bounded-variance and bounded-gradient assumptions. More recently, Cheng et al. (2025) develop variants that combine SAM with Adagrad (Duchi et al., 2011; Ward et al., 2020) and Adam (Kingma & Ba, 2015), obtaining an $\mathcal{O}\left(\log T/T^{1/4}\right)$ rate for $\frac{1}{T}\sum_{t=0}^{T-1}\mathbb{E}\|\nabla f(x^t)\|$ in the smooth case under a certain type of growth condition.

Prior adaptive analyses, primarily developed for non-convex objectives, either have additional assumptions (Sun et al., 2024) or obtain slower rates (Cheng et al., 2025), highlighting the difficulty of the theoretical analysis for adaptive step sizes for SAM. In this work, we take inspiration from the recently introduced and highly efficient adaptive Polyak step sizes for SGD and investigate their applicability and extensions to SAM and its variants. See also Table 1 for a summary of our results and comparison with closely related works.

## 1.1. Main Contributions

Our contributions are summarized as follows.

**Polyak step sizes for Sharpness-Aware Minimization.** Motivated by the success of Polyak step sizes for subgradient methods and for SGD, and by the lack of principled Polyak-type adaptivity for SAM, we propose Polyak-inspired schedulers for USAM. Starting from the USAM update, we show how ideas from the Polyak step size litera-

ture can be adapted to the sharpness-aware setting, yielding a closed-form and fully adaptive step size expressed only in terms of quantities available at the current iteration, namely the loss and the gradient evaluated at the perturbed point. When the sharpness radius is set to zero, our schedulers reduce to the classical Polyak step size and its stochastic analogue SPS$_{\max}$ from Loizou et al. (2021), thereby unifying standard and sharpness-aware gradient methods within a single framework. To the best of our knowledge, this is the first work to connect stochastic Polyak step sizes with SAM-type updates, providing a new route to adaptive sharpness-aware optimization.

**Deterministic convergence guarantees.** For deterministic (full-batch) USAM equipped with the proposed Polyak scheduler, we establish non-asymptotic convergence guarantees under standard smoothness and (strong) convexity assumptions. In particular, we prove linear convergence in squared distance $\|x^t - x^*\|^2$ for $\mu$-strongly convex objectives, matching the known rate for GD with the Polyak step size, and a sublinear $\mathcal{O}(1/T)$ rate for general convex objectives, with constants that explicitly quantify the effect of the sharpness radius. Furthermore, we relax the constant-radius condition by allowing a non-increasing radius schedule $(\rho_t)_{t\ge0}$ with $\rho_t \to 0$, and show that USAM with the Polyak scheduler still ensures $\|\nabla f(x^t)\| \to 0$ for deterministic convex objectives.

**Stochastic convergence guarantees.** In the finite-sum setting, we show that USAM with the proposed Stochastic Polyak scheduler converges up to a neighborhood of the solution of (2). Concretely, for $\mu$-strongly convex objectives we prove linear convergence in expected squared distance $\mathbb{E}\|x^t - x^*\|^2$ to a neighborhood whose size is quantified by a certain variance measure, while for general convex objectives we obtain sublinear bounds on $\mathbb{E}[f(\overline{x}^T) - f^*]$. These guarantees recover the deterministic rates when $S_t = [n]$. This provides the first convergence theory for adaptive USAM based on Polyak-type step sizes, and it avoids extra conditions commonly imposed in prior analyses of adaptive SAM-type methods, such as bounded variance, bounded gradients, or growth conditions. Moreover, as a side results of our theory, we show that the neighborhood of convergence disappears in interpolated regimes and we derive a novel convergence analysis for constant step size USAM in the stochastic regime, again without requiring any extra assumptions.

**Numerical evaluation on deep learning benchmarks.** We complement our theoretical developments with an empirical study of our Polyak schedulers for SAM on standard image classification benchmarks. Using ResNet architectures on CIFAR-10 and CIFAR-100, we compare the proposed methods against tuned SAM and SAM with cosine annealing, examining generalization performance and robustness to hyperparameter choices. We also include synthetic experiments that empirically verify the convergence guarantees predicted by our theory. Our results indicate that the Polyak schedulers can significantly reduce the learning-rate tuning burden in SAM while maintaining or improving test accuracy, supporting the practical relevance of our approach for deep learning applications.

## 2. Polyak Schedulers for USAM

This section reviews previous Polyak step size rules and introduces their counterparts for USAM. We first recall the classical Polyak step size (PS) and its stochastic analogue (SPS$_{\max}$). We then derive Polyak-style learning rates for deterministic and stochastic USAM using analogous techniques, and summarize basic properties that will be used in the convergence analysis.

### 2.1. Polyak Step Sizes: Background

The Polyak step size, introduced in Polyak (1969), is a classical adaptive choice for gradient descent (GD), $x^{t+1} = x^t - \gamma_t \nabla f(x^t)$, in convex optimization. It is given by

$$\gamma_t = \frac{f(x^t) - f^*}{\|\nabla f(x^t)\|^2}, \qquad \text{(GD-PS)}$$

and arises naturally by selecting $\gamma_t$ to minimize an upper bound on $\|x^{t+1} - x^*\|^2$ in the standard GD analysis. Beyond its original context, Polyak-type rules have also been used to analyze deterministic sub-gradient methods under various assumptions, often yielding favorable convergence guarantees (Boyd et al., 2003; Davis et al., 2018; Hazan & Kakade, 2019). While GD-PS depends on the optimal value $f^* = f(x^*)$, this quantity (or a tight lower bound) is available in several applications, such as feasibility over convex sets and positive semidefinite matrix completion (Boyd et al., 2003).

Motivated by these guarantees, Loizou et al. (2021) proposed a stochastic adaptation of Polyak's rule for SGD, $x^{t+1} = x^t - \gamma_t \nabla f_{S_t}(x^t)$. Their stochastic Polyak step size (SPS$_{\max}$) retains the main appeal of GD-PS, namely reduced dependence on problem parameters such as smoothness or strong convexity constants, while achieving rates comparable to standard SGD and showing strong empirical performance in over-parameterized regimes. More specifically, Loizou et al. (2021) proposed the SPS$_{\max}$ given by:

$$\gamma_t = \min\left\{\frac{f_{S_t}(x^t) - \ell_{S_t}^*}{\|\nabla f_{S_t}(x^t)\|^2}, \gamma_b\right\}. \qquad \text{(SPS}_{\max})$$

Here, $\ell_{S_t}^*$ is any lower bound on $f_{S_t}$. Note that in most learning problems where the losses are non-negative, one may simply take $\ell_{S_t}^* = 0$.[1] The cap $\gamma_b > 0$ prevents excessively large steps and is typically used to ensure stability and convergence to a neighborhood in the stochastic setting.

### 2.2. Polyak Scheduler for USAM

Let us now derive a Polyak-type step size tailored to USAM in the deterministic setting. The construction follows the classical GD-PS derivation: we upper bound the quantity $\|x^t - x^*\|^2$ and select $\gamma_t$ to minimize this bound. We also show that, under standard smoothness and convexity assumptions and for sufficiently small sharpness radius $\rho$, the resulting step size is automatically non-negative.

Expanding the squared distance yields:

$$\|x^{t+1} - x^*\|^2 - \|x^t - x^*\|^2$$
$$= -2\gamma_t\langle\nabla f(e^t), x^t - x^*\rangle + \gamma_t^2\|\nabla f(e^t)\|^2$$
$$= -2\gamma_t\left(\langle\nabla f(e^t), e^t - x^*\rangle - \langle\nabla f(e^t), e^t - x^t\rangle\right) + \gamma_t^2\|\nabla f(e^t)\|^2$$
$$\leq -2\gamma_t\left(f(e^t) - f^* - \langle\nabla f(e^t), e^t - x^t\rangle\right) + \gamma_t^2\|\nabla f(e^t)\|^2$$
$$= -2\gamma_t\left(f(e^t) - f^* - \rho_t\langle\nabla f(e^t), \nabla f(x^t)\rangle\right) + \gamma_t^2\|\nabla f(e^t)\|^2,$$

where the inequality follows from the fact that $f$ is convex and the last equality follows from the USAM update rule. Minimizing the resulting quadratic upper bound over $\gamma_t \geq 0$

---

[1]The original formulation in Loizou et al. (2021) uses the mini-batch optimal value $f_{S_t}^* = \inf_{x \in \mathbb{R}^d} f_{S_t}(x)$. Following Orvieto et al. (2022), one can replace it with a valid lower bound $\ell_{S_t}^*$ without changing the essence of the convergence guarantees.

*Table 2.* Correspondence between Polyak step sizes for SGD and USAM. Here $f^* = \inf_{x \in \mathbb{R}^d} f(x)$, and $\ell^*_{S_t}$ is any lower bound for the mini-batch objective $f_{S_t}$ (typically $\ell^*_{S_t} = 0$ for non-negative losses).

| Setting | SGD | USAM |
|---|---|---|
| Deterministic | $\gamma_t = \dfrac{f(x^t) - f^*}{\|\nabla f(x^t)\|^2}$ | $\gamma_t = \dfrac{f(e^t) - f^* - \rho_t \langle \nabla f(e^t), \nabla f(x^t) \rangle}{\|\nabla f(e^t)\|^2}$ |
| Stochastic | $\gamma_t = \min\left\{ \dfrac{f_{S_t}(x^t) - \ell^*_{S_t}}{\|\nabla f_{S_t}(x^t)\|^2}, \gamma_b \right\}$ | $\gamma_t = \min\left\{ \dfrac{f_{S_t}(e^t) - \ell^*_{S_t} - \rho_t \langle \nabla f_{S_t}(e^t), \nabla f_{S_t}(x^t) \rangle}{\|\nabla f_{S_t}(e^t)\|^2}, \gamma_b \right\}$ |

gives $\gamma_t = \frac{\left[ f(e^t) - f^* - \rho_t \langle \nabla f(e^t), \nabla f(x^t) \rangle \right]_+}{\|\nabla f(e^t)\|^2}$, where $[z]_+ = \max\{z, 0\}$. The ReLU safeguard guarantees $\gamma_t \geq 0$ by construction. In the smooth convex regime of interest, it is in fact unnecessary: for sufficiently small $\rho_t$, the numerator is automatically non-negative, and the ReLU can be dropped without changing the analysis.

**Proposition 2.1** (Non-negativity and lower bound). Let $f$ be convex and $L$-smooth. If $\rho_t \leq 1/L$, then

$$\gamma_t \geq \frac{1 - L\rho_t}{2L(1 + L\rho_t)} \geq 0. \qquad (3)$$

In particular, in this regime the ReLU safeguard is redundant, and the Polyak Scheduler for the deterministic USAM can be written as

$$\gamma_t = \frac{f(e^t) - f^* - \rho_t \langle \nabla f(e^t), \nabla f(x^t) \rangle}{\|\nabla f(e^t)\|^2}.$$
(Polyak Scheduler)

The proof can be found in Section C. Furthermore, we have the following descent property, which will be used repeatedly in the convergence analysis.

**Proposition 2.2** (Descent property of Polyak Scheduler). Let $f$ be convex and $L$-smooth, and suppose $\rho_t \leq 1/L$. Then the iterates generated by deterministic USAM with Polyak Scheduler satisfy, for all $t \geq 0$,

$$\|x^{t+1} - x^*\|^2 \leq \|x^t - x^*\|^2 - \tfrac{(1 - L\rho_t)^2}{2L} (f(x^t) - f^*). \qquad (4)$$

Notably, the sequence $\{\|x^t - x^*\|\}_{t \geq 0}$ is non-increasing.

### 2.3. Stochastic Polyak Scheduler for USAM

We now extend the above construction to the stochastic setting, in direct analogy with the passage from GD-PS to SPS$_{\max}$. Let $S_t$ be a random mini-batch of fixed size $\tau$. We propose the following capped Polyak-type step size for stochastic USAM:

$$\gamma_t = \min\left\{ \frac{f_{S_t}(e^t) - \ell^*_{S_t} - \rho_t \langle \nabla f_{S_t}(e^t), \nabla f_{S_t}(x^t) \rangle}{\|\nabla f_{S_t}(e^t)\|^2}, \gamma_b \right\}.$$
(Stochastic Polyak Scheduler)

Here, the parameter $\gamma_b > 0$ has the same purpose as in the original SPS$_{\max}$, and it is a bound that restricts Stochastic Polyak Scheduler from being too big and is essential to ensure convergence to a neighborhood of the solution. Additionally, $\ell^*_{S_t}$ is any lower bound of $f_{S_t}$ (typically $\ell^*_{S_t} = 0$ for non-negative losses), mirroring the relaxation for SPS$_{\max}$.

Moreover, if each component loss $f_i$ is $L_i$-smooth and we set $L_{\max} = \max_{i \in [n]} L_i$, then, defining $L_{S_t} = \frac{1}{\tau} \sum_{i \in S_t} L_i \leq L_{\max}$, the same smoothness argument as in Proposition 2.1 yields that whenever $\rho_t \leq 1/L_{\max}$, we have $\frac{f_{S_t}(e^t) - \ell^*_{S_t} - \rho_t \langle \nabla f_{S_t}(e^t), \nabla f_{S_t}(x^t) \rangle}{\|\nabla f_{S_t}(e^t)\|^2} \geq \frac{1 - L_{S_t}\rho_t}{2L_{S_t}(1 + L_{S_t}\rho_t)} \geq \frac{1 - L_{\max}\rho_t}{2L_{\max}(1 + L_{\max}\rho_t)}$, and therefore

$$\gamma_b \geq \gamma_t \geq \min\left\{ \frac{1 - L_{\max}\rho_t}{2L_{\max}(1 + L_{\max}\rho_t)}, \gamma_b \right\}. \qquad (5)$$

As a final remark, observe that in both the deterministic and stochastic settings, setting $\rho_t = 0$ reduces Polyak Scheduler and Stochastic Polyak Scheduler to the classical Polyak rules GD-PS and SPS$_{\max}$, respectively; see Table 2 for a summary of this correspondence.

## 3. Convergence Analysis

This section presents convergence guarantees for deterministic and stochastic USAM equipped with Polyak Scheduler and Stochastic Polyak Scheduler. Complete proofs are deferred to Appendix D.

### 3.1. Deterministic

We first consider the deterministic (full-batch) setting.

**Theorem 3.1** (Strongly convex case). Let $f$ be $\mu$-strongly convex and $L$-smooth. Suppose that $\rho_t = \rho \leq \frac{1}{L}$. Then the iterates generated by deterministic USAM with Polyak Scheduler satisfy, for all $t \geq 0$,

$$\|x^t - x^*\|^2 \leq \left( 1 - \frac{\mu(1 - L\rho)^2}{4L} \right)^t \|x^0 - x^*\|^2.$$

Theorem 3.1 establishes a linear convergence rate in squared distance to the minimizer. When $\rho = 0$, deterministic USAM reduces to GD and Polyak Scheduler reduces to the

classical Polyak step size GD-PS. In this case, our theorem's contraction factor becomes $1 - \mu/(4L)$, which matches the standard $\mathcal{O}(\mu/L)$ rate for GD with the Polyak step size.

We next turn to the general convex case.

**Theorem 3.2** (Convex case). Let $f$ be convex and $L$-smooth. Suppose that $\rho_t = \rho \leq \frac{1}{L}$. Then the iterates generated by deterministic USAM with Polyak Scheduler satisfy, for all $T \geq 1$,

$$f(\overline{x}^T) - f^* \leq \frac{2L\|x^0 - x^*\|^2}{T(1 - L\rho)^2},$$

where $\overline{x}^T = \frac{1}{T} \sum_{t=0}^{T-1} x^t$ is the Cesaro average.

Similarly to the previous theorem, when $\rho = 0$ in Theorem 3.2 the rate reduces to $\frac{2L\|x^0-x^*\|^2}{T}$, matching the standard $O(1/T)$ rate for GD with GD-PS.

**Remark 3.3** (Effect of the radius in the deterministic bounds). The right-hand side in Theorem 3.2 is increasing in $\rho \in [0, 1/L)$, and the same is true for the bound in Theorem 3.1. Thus, from the perspective of these optimization guarantees, the sharpest bound is obtained at $\rho = 0$, in which case USAM reduces to gradient descent and our rates recover the standard ones for GD with the Polyak step size. This observation should not be interpreted as a statement against sharpness-aware updates: in SAM-type methods, choosing $\rho > 0$ is primarily motivated by generalization considerations rather than by worst-case convex optimization rates.

The deterministic results above use a constant radius $\rho$ that depends on the smoothness constant through the condition $\rho \leq 1/L$. A natural way to alleviate this restriction is to allow a decreasing sequence of radii. The following theorem shows that, as long as $\rho_t \downarrow 0$, the method drives the gradient norm to zero.

**Theorem 3.4** (Decreasing radius implies vanishing gradients). Let $f$ be convex and $L$-smooth. Consider deterministic USAM with Polyak Scheduler, where the radii $(\rho_t)_{t \geq 0}$ satisfy $\rho_t \downarrow 0$, meaning that $\rho_t \geq 0$, $\rho_{t+1} \leq \rho_t$ for all $t$, and $\rho_t \to 0$. Then $\sum_{t=0}^{\infty} (f(x^t) - f^*) < \infty$, and consequently $f(x^t) \to f^*$ as $t \to \infty$.

### 3.2. Stochastic

We now consider the stochastic finite-sum problem (2). The results below show that stochastic USAM with Stochastic Polyak Scheduler converges up to a neighborhood whose size is quantified by the following variance-type measure.

**Variance measure.** To quantify the limiting neighborhood, we define

$$\sigma^2 := \mathbb{E}_{S_t}\left[f_{S_t}(x^*) - \ell_{S_t}^*\right] = f(x^*) - \mathbb{E}_{S_t}[\ell_{S_t}^*], \quad (6)$$

where $\ell_{S_t}^*$ is any lower bound on the mini-batch objective $f_{S_t}$. This notion is standard in the stochastic Polyak step size literature (Loizou et al., 2021; Wang et al., 2023; Zhang et al., 2025). Since each $f_i$ is lower bounded, $\sigma^2 < \infty$. We say that (2) is *interpolated* if $\sigma^2 = 0$, meaning that there exists $x^* \in X^*$ such that $f(x^*) = f_{S_t}(x^*) = \ell_{S_t}^*$. This condition is satisfied in many over-parameterized learning settings; see, e.g., Liang & Rakhlin (2020) for nonparametric regression and Ma et al. (2018); Zhang et al. (2021) for discussions related to deep networks.

Now we are ready for stochastic statements. We start with the strongly convex case.

**Theorem 3.5** (Strongly convex case). Let each $f_i$ be convex and $L_i$-smooth, and set $L_{\max} = \max_{i \in [n]} L_i$. Suppose that $f$ is $\mu$-strongly convex and that $\rho_t = \rho \leq \frac{1}{L_{\max}}$. Let $\alpha = (1 - L_{\max}\rho)^2 \min\left\{\frac{1}{2L_{\max}}, \gamma_b\right\}$. Then the iterates of USAM with Stochastic Polyak Scheduler satisfy, for all $t \geq 0$,

$$\mathbb{E}\|x^t - x^*\|^2 \leq \left(1 - \frac{\mu\alpha}{2}\right)^t \|x^0 - x^*\|^2 + \frac{2(2\gamma_b - \alpha)}{\mu\alpha}\sigma^2.$$

Theorem 3.5 shows that, for strongly convex objectives, USAM with Stochastic Polyak Scheduler converges linearly to a neighborhood whose size is controlled by $\sigma^2$. Notably, the guarantee does not require additional assumptions such as bounded gradients or growth conditions. When $\rho = 0$, stochastic USAM reduces to SGD and Stochastic Polyak Scheduler reduces to $\text{SPS}_{\max}$. In this case, the bound of Theorem 3.5 matches (up to constants) the $\text{SPS}_{\max}$ guarantee of Loizou et al. (2021) for SGD on smooth strongly convex objectives.

When interpolation holds ($\sigma^2 = 0$), the neighborhood term vanishes. In this case, the step size cap is not needed and one may take $\gamma_b = \infty$.

**Corollary 3.6** (Interpolation). Assume interpolation ($\sigma^2 = 0$) and let the assumptions of Theorem 3.5 hold. Then the iterates of USAM with Stochastic Polyak Scheduler and $\gamma_b = \infty$ satisfy, for all $t \geq 0$,

$$\mathbb{E}\|x^t - x^*\|^2 \leq \left(1 - \frac{\mu(1 - L_{\max}\rho)^2}{4L_{\max}}\right)^t \|x^0 - x^*\|^2.$$

Another consequence of Theorem 3.5 is a convergence guarantee for constant step size USAM.

**Corollary 3.7** (Constant step size). Let the assumptions of Theorem 3.5 hold and suppose $\rho_t = \rho$. If $\gamma_b \leq$

$\frac{1-L_{\max}\rho}{2L_{\max}(1+L_{\max}\rho)}$, then Stochastic Polyak Scheduler yields $\gamma_t \equiv \gamma_b$ and thus it becomes USAM with constant step size $\gamma = \gamma_b$. Moreover, the iterates satisfy, for all $t \geq 0$,

$$\mathbb{E}\|x^t - x^*\|^2 \leq \left(1 - \frac{\mu(1-L_{\max}\rho)^2\gamma}{2}\right)^t \|x^0 - x^*\|^2 + \frac{2(2 - (1-L_{\max}\rho)^2)}{\mu(1-L_{\max}\rho)^2}\sigma^2.$$

Related guarantees for USAM have been obtained in Andriushchenko & Flammarion (2022) and Oikonomou & Loizou (2025b). In Andriushchenko & Flammarion (2022), the authors establish an $\mathcal{O}(1/T)$ rate for $\mathbb{E}[f(x^T) - f^*]$ using diminishing parameters $\gamma_t = \mathcal{O}(1/t)$ and $\rho_t = \mathcal{O}(1/\sqrt{t})$, under a bounded-variance assumption. Oikonomou & Loizou (2025b) derive linear rates with constant step sizes under the Expected Residual condition. In contrast, our analysis provides linear convergence in $\mathbb{E}\|x^t - x^*\|^2$ without additional assumptions beyond smoothness and strong convexity. Moreover, a direct comparison shows that the constant step size permitted by Corollary 3.7 is strictly larger than the corresponding step sizes in these works.

We finally state the counterpart of Theorem 3.2 for the general convex stochastic setting.

**Theorem 3.8** (Convex case). Let each $f_i$ be convex and $L_i$-smooth, and set $L_{\max} = \max_{i\in[n]} L_i$. Suppose that $\rho_t = \rho \leq \frac{1}{L_{\max}}$. Let $\alpha = (1-L_{\max}\rho)^2 \min\left\{\frac{1}{2L_{\max}}, \gamma_b\right\}$. Then the iterates of USAM with Stochastic Polyak Scheduler satisfy, for all $T \geq 1$,

$$\mathbb{E}[f(\overline{x}^T) - f^*] \leq \frac{\|x^0 - x^*\|^2}{\alpha T} + \frac{2\gamma_b - \alpha}{\alpha}\sigma^2,$$

Analogous to the strongly convex case, several specializations of Theorem 3.8 are immediate. When $\rho = 0$, the bound matches (up to constants) the SPSmax guarantee of Loizou et al. (2021) for SGD on smooth convex objectives. When interpolation holds ($\sigma^2 = 0$), the neighborhood term vanishes and the method converges to the optimum. In direct analogy with Corollary 3.7, choosing $\gamma_b \leq \frac{1-L_{\max}\rho}{2L_{\max}(1+L_{\max}\rho)}$ further yields a convergence guarantee for constant-step USAM in the convex regime. Finally, both Theorems 3.5 and 3.8 recover their deterministic counterparts when $S_t = [n]$ and $\ell_{S_t}^* = f_{S_t}^*$.

## 4. Numerical experiments

This section evaluates the proposed Polyak schedulers on synthetic problems and deep-learning benchmarks. We first verify on synthetic ridge regression that USAM with Polyak Scheduler exhibits the linear trends predicted by our theory.

We then assess Stochastic Polyak Scheduler on CIFAR-10/100 with ResNets, comparing to tuned constant learning rates and cosine annealing. Finally, although our theory focuses on USAM, we also extend the Polyak schedulers to SAM and report corresponding experiments. We provide the code for all of our experiments at https://github.com/dimitris-oik/sam_sps.

### 4.1. Verification of the theory

We empirically validate our theoretical guarantees on synthetic strongly convex problems, using USAM with Polyak Scheduler. We compare our step sizes against constant step size USAM baselines that come with convergence guarantees in the literature.

We consider a regularized Ridge regression objective of the form $f(x) = \frac{1}{2n}\sum_{i=1}^n (A[i,:]x - b_i)^2 + \frac{\lambda_r}{2}\|x\|^2$, where $A_i \in \mathbb{R}^d$ are the rows of a matrix $A \in \mathbb{R}^{n \times d}$ and $b \in \mathbb{R}^n$ and $\lambda_r$ is the regularization constant. In our experiments we set $n = d = 100$ and generate $A$ using the procedure of Lenard & Minkoff (1984) so that $\kappa(A) = 10$, ensuring strong convexity. In the deterministic case we set $\lambda_r = 0$ and we generate a consistent linear system by first sampling $x^* \in \mathbb{R}^d$ and then setting $b = Ax^*$, which implies $f^* = 0$. In the stochastic case, we set $\lambda_r = 10^{-3}$ and we use an approximation of $f^*$ in closed form.

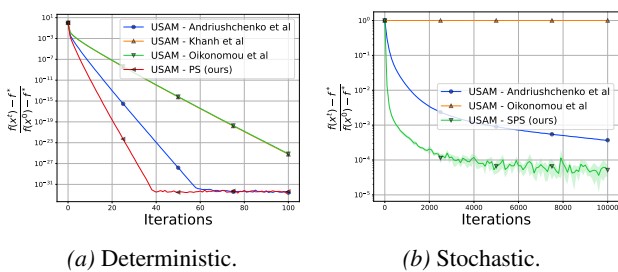

*(a)* Deterministic.  *(b)* Stochastic.

*Figure 1.* Synthetic Ridge Regression.

**Deterministic regime.** We compare USAM with Polyak Scheduler against representative constant step size USAM baselines with deterministic convergence guarantees. Specifically, Khanh et al. (2024) assume $\rho \in [0, 1/L)$ and $\gamma \in [0, 4/(9L))$, Andriushchenko & Flammarion (2022) assume $\rho \in [0, 1/L)$ and $\gamma \in [0, 1/L)$, and Oikonomou & Loizou (2025b) provide a linear convergence rate for $\rho \in [0, 1/(3L))$ and $\gamma \in \left[0, \frac{1-3L\rho}{L(2L^2\rho^2+1)}\right)$. For each baseline, we choose the pair $(\rho, \gamma)$ according to the following rule: when the guarantee requires an interval open on the right (e.g., $\rho \in [0, 1/L)$), we use the midpoint, while for closed intervals we take the maximum possible value. The resulting curves are shown in Figure 1a. All methods exhibit the expected linear convergence, with USAM and Polyak Scheduler converging fastest in this test.

**Stochastic regime.** We compare against the following USAM step sizes with stochastic guarantees: Andriushchenko & Flammarion (2022) they provide a $\mathcal{O}\left(1/T^2\right)$ rate for $\rho_t = \sqrt{\gamma_t/L_{\max}}$ and $\gamma_t = \min\left\{\frac{8t+4}{3\mu(t+1)^2}, \frac{1}{2L_{\max}}\right\}$. In Oikonomou & Loizou (2025b) they provide a linear rate for $\rho < \frac{\mu}{L_{\max}(\mu+2L_{\max})}$ and $\gamma < \frac{\mu - L_{\max}\rho(\mu+2L_{\max})}{2L_{\max}^2(2L_{\max}^2\rho^2+1)}$. We repeat each stochastic experiment 5 times, using mini-batches $S_t$ drawn uniformly at random from all subsets of size $\tau = 10$ and independently across iterations, and report the mean $\pm$ one standard deviation. The results in Figure 1b show that Stochastic Polyak Scheduler converges to a neighborhood of the solution, consistent with our stochastic convergence guarantees in Section 3.2.

**Comparison with other adaptive SAM optimizers.** On the same Ridge regression problems as above ($n = d = 100$, $\kappa(\boldsymbol{A}) = 10$, consistent system, $f^* = 0$), we further benchmark our Polyak Scheduler against several recently proposed deterministic adaptive SAM variants: AdaSAM (Sun et al., 2024), which equips SAM with an AMSGrad-style learning rate and momentum; the three algorithms of Cheng et al. (2025), which adapt both the learning rate and the perturbation radius using AdaGrad-Norm (LightSAM-I), AdaGrad (LightSAM-II), and Adam (LightSAM-III) updates; and SA-SAM (Naganuma et al., 2024), which sets the learning rate from an adaptive local-smoothness estimate. For each baseline, we sweep its scalar hyperparameters over a small grid and report the best-performing configuration; our Polyak Scheduler runs without any tuning beyond the perturbation radius $\rho$.

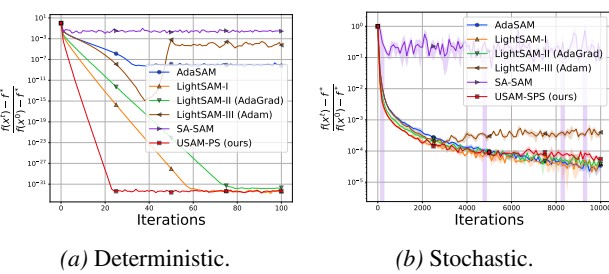

*(a)* Deterministic.    *(b)* Stochastic.

*Figure 2.* Comparison with other adaptive SAM optimizers.

The results are shown in Section 4.1. In the deterministic regime, our scheduler converges to the optimum in the fewest iterations; the AdaGrad-based LightSAM variants reach the same accuracy but require noticeably more iterations, while the remaining baselines stall further from the optimum. In the stochastic regime, our scheduler is competitive with the strongest baselines, converging at a comparable rate to a similar neighborhood of the solution. Beyond this empirical performance, our scheduler is the only adaptive method in the comparison that comes with convergence

guarantees for smooth, strongly convex objectives.

### 4.2. USAM-SPS on DNNs

In this subsection, we evaluate the practical behavior of the proposed Stochastic Polyak Scheduler on standard image-classification benchmarks. We train ResNet-20 and ResNet-32 models (He et al., 2016) on CIFAR-10 and CIFAR-100 (Krizhevsky et al., 2009) for 100 epochs using cross-entropy loss and mini-batches of size $\tau = 128$. We apply standard data augmentation (random crop and random horizontal flip) followed by normalization (DeVries, 2017). Unless stated otherwise, we report results with weight decay wd $= 5 \cdot 10^{-4}$ (additional results, including wd $= 0.0$, are provided in Appendix E). All experiments are run on NVIDIA RTX 6000 Ada GPUs. Each configuration is repeated over three seeds, and we report the best test accuracy over epochs (mean $\pm$ one standard deviation).

**Baselines and tuning protocol.** We compare USAM equipped with the proposed Stochastic Polyak Scheduler against two widely used learning-rate baselines: (i) *constant step size* USAM with a tuned learning rate $\gamma$, and (ii) USAM combined with cosine annealing (Loshchilov & Hutter, 2017). For constant step size USAM, we tune $\gamma$ separately for each radius $\rho \in \{0.1, 0.2, 0.3, 0.4\}$ via a grid search (we use $\gamma \in \{10^{-3}, 10^{-2}, 10^{-1}\}$, and in our runs $\gamma = 0.1$ is consistently best across all radii). For cosine annealing, we use the PyTorch (Paszke et al., 2019) implementation given by:

$$\gamma_{t+1} = \gamma_{\min} + (\gamma_t - \gamma_{\min}) \cdot \frac{1 + \cos\left(\frac{(t+1)\pi}{T}\right)}{1 + \cos\left(\frac{t\pi}{T}\right)},$$
(Cosine Annealing)

where $T$ is the number of epochs and $\gamma_{\min}$ is a hyperparameter. We tune the minimum learning rate $\gamma_{\min}$ over the same grid $\gamma_{\min} \in \{10^{-3}, 10^{-2}, 10^{-1}\}$ across each radius. Finally, for Stochastic Polyak Scheduler we set the mini-batch lower bound to $\ell_{S_t}^* = 0$ and $\gamma_b = 1.0$.

Table 3 summarizes our CIFAR-100 results with ResNet-32 (additional architectures/datasets are deferred to Appendix E). Overall, USAM equipped with Stochastic Polyak Scheduler matches the performance of tuned baselines and typically outperforms both a constant learning rate and Cosine Annealing. In addition, as the sharpness radius $\rho$ increases, Cosine Annealing exhibits a significant fall-off in accuracy, whereas Stochastic Polyak Scheduler maintains substantially higher performance. Finally, we emphasize that, in contrast to Cosine Annealing, Stochastic Polyak Scheduler is a principled choice that follows directly from our theoretical development.

*Table 3.* CIFAR-100 test accuracy (%, mean $\pm$ std over 3 seeds) for ResNet-32 trained with USAM under three learning-rate schedules.

| | Constant USAM | USAM with Cosine Annealing | USAM with Stochastic Polyak Scheduler |
|---|---|---|---|
| $\rho = 0.1$ | $90.56_{\pm 0.18}$ | $90.01_{\pm 0.32}$ | $\mathbf{91.81_{\pm 0.04}}$ |
| $\rho = 0.2$ | $90.45_{\pm 0.34}$ | $88.77_{\pm 0.26}$ | $\mathbf{92.23_{\pm 0.22}}$ |
| $\rho = 0.3$ | $90.25_{\pm 0.10}$ | $88.05_{\pm 0.23}$ | $\mathbf{92.24_{\pm 0.30}}$ |
| $\rho = 0.4$ | $89.56_{\pm 0.07}$ | $86.52_{\pm 0.04}$ | $\mathbf{92.01_{\pm 0.12}}$ |

*Table 4.* CIFAR-100 test accuracy (%, mean $\pm$ std over 3 seeds) for ResNet-32 trained with SAM under three learning-rate schedules.

| | Constant SAM | SAM with Cosine Annealing | SAM with Stochastic Polyak Scheduler |
|---|---|---|---|
| $\rho = 0.1$ | $90.17_{\pm 0.11}$ | $90.49_{\pm 0.02}$ | $\mathbf{91.61_{\pm 0.12}}$ |
| $\rho = 0.2$ | $90.53_{\pm 0.02}$ | $89.03_{\pm 0.13}$ | $\mathbf{92.24_{\pm 0.07}}$ |
| $\rho = 0.3$ | $89.61_{\pm 0.10}$ | $87.05_{\pm 0.24}$ | $\mathbf{91.70_{\pm 0.15}}$ |
| $\rho = 0.4$ | $88.64_{\pm 0.13}$ | $84.61_{\pm 0.34}$ | $\mathbf{90.79_{\pm 0.16}}$ |

### 4.3. SAM-SPS on DNNs

So far, our theory and step size construction have focused on the unnormalized update USAM. In this subsection, we show that the same Polyak-style principle also yields a natural adaptive step size for normalized SAM, and we report corresponding experiments on the same benchmarks.

SAM differs from USAM only in the normalization of the perturbation step, but the same Polyak-style upper-bound argument applies. Specializing the derivation of Sections 2.2 and 2.3 to the SAM update yields the following adaptive step sizes. In the deterministic (full-batch) setting we get $\gamma_t = \frac{\left[f(e^t) - f^* - \langle \nabla f(e^t), e^t - x^t \rangle\right]_+}{\|\nabla f(e^t)\|^2}$ and in the stochastic mini-batch setting, we use:

$$\gamma_t = \min\left\{\frac{\left[f_{S_t}(e^t) - \ell^*_{S_t} - \langle \nabla f_{S_t}(e^t), e^t - x^t \rangle\right]_+}{\|\nabla f_{S_t}(e^t)\|^2}, \gamma_b\right\}.$$
(Stochastic Polyak Scheduler)

In contrast to the USAM case (cf. Proposition 2.1), our non-negativity argument does not directly apply to SAM, hence we retain the safeguard $[\cdot]_+$. However, we observe that this safeguard is rarely active in our DNN experiments.

We use the same setup as in Section 4.2 (architectures, datasets, etc). We compare: (i) SAM with a tuned constant learning rate, (ii) SAM with Cosine Annealing, and (iii) SAM with Stochastic Polyak Scheduler with $\ell^*_{S_t} = 0$ and $\gamma_b = 1.0$. Table 4 reports the ResNet-32 performance on CIFAR-100 (additional experiments are deferred to Appendix E). The same trends observed for USAM persist for the normalized update: equipping SAM with our adaptive rule Stochastic Polyak Scheduler consistently improves upon the constant-step and Cosine Annealing variants. Notably, the advantage becomes more pronounced at larger sharpness radii $\rho$, where Cosine Annealing deteriorates while Stochastic Polyak Scheduler remains comparatively stable.

## 5. Conclusion

In this work, we introduced Polyak schedulers for unnormalized Sharpness-Aware Minimization (USAM), yielding closed-form and fully adaptive learning rates computed from quantities available at each iteration. We proved linear convergence for strongly convex objectives and a sublinear $\mathcal{O}(1/T)$ convergence rate for convex objectives in the deterministic case. In the stochastic setting, we establish analogous convergence guarantees up to a neighborhood of the optimum. These results provide a principled approach to reducing learning-rate tuning in SAM-type methods while preserving theoretical guarantees.

Several directions remain open. On the theoretical side, an important next step is to extend the analysis beyond convexity, for instance, to Polyak-Lojasiewicz objectives (Karimi et al., 2016), weakly convex, or more general non-convex objectives, while retaining Polyak-style adaptivity. On the algorithmic side, developing principled strategies for selecting or scheduling the sharpness radius $\rho_t$ is a natural direction, as this parameter plays a central role in balancing optimization performance and generalization. More broadly, applying Polyak-scheduled SAM methods to large-scale training regimes, including large language models (LLMs), could further clarify the practical benefits of adaptive, tuning-light, sharpness-aware optimization.

## Impact Statement

This paper presents work whose goal is to advance the field of Machine Learning. There are many potential societal consequences of our work, none of which we feel must be specifically highlighted here.

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

# Supplementary Material

The Supplementary Material is organized as follows. Section A reviews additional related work on adaptive step-size methods. In Section B, we collect basic definitions and the auxiliary lemmas used throughout. Section C contains the proofs of the basic properties of Polyak Scheduler. Section D provides the proofs of the main theoretical guarantees. Finally, Section E includes additional experimental results.

## A. Further Related Work on Adaptive Methods

### A.1. On Polyak-type Step Sizes

A growing literature aims to eliminate learning-rate tuning by setting steps from readily available quantities (typically the sampled loss value and a gradient norm), while retaining convergence guarantees. In this line, Gower et al. (2022) systematize SPS-type rules for SGD, reinterpret them through a Passive–Aggressive/slack-variable lens, and propose "slack" variants that stabilize Polyak-style updates beyond ideal interpolation regimes. Building on known drawbacks of vanilla SPS in non-interpolated settings, Orvieto et al. (2022) analyzes the dynamics induced by stochastic Polyak step sizes, identifies biases that can prevent exact convergence, and introduces truly adaptive variants (e.g., decreasing-step constructions) that provably converge to the minimizer without requiring problem parameters. Complementarily, Garrigos et al. (2023) studies function-value-driven Polyak variants by formalizing an idealized positive SPS scheme and proposing "Function Value Learning", which learns (rather than assumes) the per-sample optimal loss values needed by idealized Polyak rules. To make Polyak-style adaptation compatible with composite objectives, Schaipp et al. (2023) develops a stochastic proximal Polyak step size that incorporates regularization via proximal steps, addressing the practical need for robustness when losses are paired with non-smooth terms.

On the momentum side, Schaipp et al. (2024) propose MoMo, which uses momentum-based models of sampled losses/gradients (together with truncation via lower bounds) to generate Polyak-type adaptive learning rates that plug naturally into momentum methods. Furthermore, Gower et al. (2025) analyze an *idealized* stochastic Polyak method for momentum, yielding general convergence guarantees and applications including black-box model distillation. More directly, Oikonomou & Loizou (2025c) design and analyze Polyak-type step sizes for stochastic heavy-ball momentum, providing guarantees both to neighborhoods (without interpolation) and to the exact minimizer (via adaptive/decreasing variants) without prior knowledge of smoothness/strong convexity constants.

For non-smooth optimization, Oikonomou & Loizou (2025a) introduce safeguarded SPS rules for stochastic (sub)gradient methods that avoid failure modes from small/vanishing (sub)gradients and prove robust performance guarantees in non-smooth settings. Beyond standard SGD, D'Orazio et al. (2023) generalize Polyak-style adaptation to mirror descent via a mirror stochastic Polyak step size, deriving convergence results and adaptive variants in non-Euclidean geometries. At the distributed scale, Mukherjee et al. (2024) study locally adaptive learning rates in federated learning, showing how Polyak/SPS-flavored local adaptation can improve convergence under heterogeneity. Finally, Orvieto & Xiao (2024) propose a related tuning-free adaptive stochastic gradient method based on non-negative Gauss–Newton step sizes, offering another principled route to function-value-informed step selection.

### A.2. Adaptive methods beyond Polyak step sizes

A separate and widely used family of adaptive methods sets the step size based on running statistics of past gradients rather than on the loss value. AdaGrad (Duchi et al., 2011; Ward et al., 2020) scales the learning rate inversely by the sum of squared gradients seen so far, which is particularly effective on sparse problems. More recently, (Choudhury et al., 2024) proposes a scale-invariant variant that removes the square-root normalization from AdaGrad's denominator, yielding a simpler and computationally efficient adaptive update. RMSProp (Tieleman & Hinton, 2012) and Adam/AdamW (Kingma & Ba, 2015; Loshchilov & Hutter, 2019) replace this sum with exponentially weighted moving averages of the first and second gradient moments, and have become the de facto choice for training deep neural networks. The theoretical analysis of these methods has been refined over time, including corrections to the original Adam analysis and tighter conditions under which they match or fall short of SGD rates (Reddi et al., 2018; Défossez et al., 2022). Variance-reduced adaptive schemes provide another route to adaptivity: AI-SARAH (Shi et al., 2023) combines stochastic recursive gradient estimators with adaptive and implicit step-size choices, reducing sensitivity to manually tuned learning-rate schedules while preserving the benefits of SARAH-type variance reduction.

A more recent and conceptually distinct direction is *parameter-free* or *learning-rate-free* optimization. Defazio & Mishchenko (2023) propose D-Adaptation, an SGD-style method that maintains a running lower bound on the initial distance to the optimum, $D = \|x^0 - x^*\|$, and uses it to set the step size online, achieving the optimal non-asymptotic rate for convex Lipschitz problems without any prior knowledge of $D$. Building on this construction, Mishchenko & Defazio (2023) introduce Prodigy, which improves on D-Adaptation through a sharper estimator and weighted averaging, closing the gap to optimally tuned SGD up to logarithmic factors and performing strongly in deep-learning benchmarks. In a separate line of work, Malitsky & Mishchenko (2020) propose an adaptive gradient descent scheme that requires neither line-search nor knowledge of the smoothness constant: the step size is updated from the current and previous iterates via a local Lipschitz estimate, and the method is shown to converge on smooth convex problems and to extend naturally to the proximal and accelerated settings.

# B. Technical Preliminaries

## B.1. Basic Definitions and Lemmas

This section collects definitions and standard inequalities that are used throughout the paper. Unless stated otherwise, $\|\cdot\|$ denotes the Euclidean norm.

**Definition B.1** (Convexity). A differentiable function $f : \mathbb{R}^d \to \mathbb{R}$ is *convex* if

$$f(x) \geq f(y) + \langle \nabla f(y), x - y \rangle \qquad \forall\, x, y \in \mathbb{R}^d. \tag{7}$$

Interchanging $x$ and $y$ in (7) and adding the two inequalities we get:

$$\langle \nabla f(x) - \nabla f(y), x - y \rangle \geq 0 \qquad \forall\, x, y \in \mathbb{R}^d. \tag{8}$$

We say that $f$ is $\mu$-*strongly convex* if and only if the function $g : \mathbb{R}^d \to \mathbb{R}$ defined by

$$g(x) = f(x) - \frac{\mu}{2}\|x\|^2$$

is convex.

**Definition B.2** ($L$-smoothness). A differentiable function $f : \mathbb{R}^d \to \mathbb{R}$ is $L$-*smooth* if there exists $L > 0$ such that

$$\|\nabla f(x) - \nabla f(y)\| \leq L\|x - y\| \qquad \forall\, x, y \in \mathbb{R}^d. \tag{9}$$

Equivalently, $f$ is $L$-smooth if and only if it satisfies the descent lemma

$$f(x) \leq f(y) + \langle \nabla f(y), x - y \rangle + \frac{L}{2}\|x - y\|^2 \qquad \forall\, x, y \in \mathbb{R}^d. \tag{10}$$

The following lemmas are standard, so we omit the proofs.

**Lemma B.3** (Properties of strong convexity). If $f$ is $\mu$-strongly convex, then for all $x, y \in \mathbb{R}^d$ we have

$$\langle \nabla f(x) - \nabla f(y), x - y \rangle \geq \mu\|x - y\|^2. \tag{11}$$

Moreover, for any $x^* \in \arg\min f$ (which implies $\nabla f(x^*) = 0$) the following hold:

$$\frac{\mu}{2}\|x - x^*\|^2 \leq f(x) - f^* \leq \frac{1}{2\mu}\|\nabla f(x)\|^2, \qquad \text{(Quadratic growth / PL bound)} \tag{12}$$

$$\mu\|x - x^*\| \leq \|\nabla f(x)\|. \qquad \text{(Error bound)} \tag{13}$$

**Lemma B.4** (Properties of smoothness). If $f$ is $L$-smooth, then for all $x, y \in \mathbb{R}^d$ we have

$$\langle \nabla f(x) - \nabla f(y), x - y \rangle \leq L\|x - y\|^2. \tag{14}$$

Moreover, for any $x^* \in \arg\min f$ the following hold:

$$\frac{1}{2L}\|\nabla f(x)\|^2 \leq f(x) - f^* \leq \frac{L}{2}\|x - x^*\|^2, \tag{15}$$

$$\|\nabla f(x)\| \leq L\|x - x^*\|. \tag{16}$$

## B.2. Auxiliary lemmas for the convergence analysis

We now record several inequalities specific to the USAM dynamics that will be used repeatedly in the convergence proofs. Throughout this subsection, we consider the deterministic USAM iterates

$$e^t = x^t + \rho_t \nabla f(x^t), \tag{17}$$

$$x^{t+1} = x^t - \gamma_t \nabla f(e^t). \tag{18}$$

**Remark B.5** (On the case $\rho_t = 0$). Some proofs below divide by $\rho_t$. When $\rho_t = 0$, USAM reduces to gradient descent and we have $e^t = x^t$. In that case, all inequalities stated in the lemmas below remain valid (typically as equalities). Thus, whenever a division by $\rho_t$ appears, it should be interpreted as applying to the case $\rho_t > 0$, with the case $\rho_t = 0$ handled separately as above.

**Lemma B.6** (Convex case). Suppose that $f$ is convex. Then the iterates (17)–(18) satisfy

$$\langle \nabla f(e^t), \nabla f(x^t) \rangle \geq \|\nabla f(x^t)\|^2, \tag{19}$$

$$\rho_t \langle \nabla f(e^t), \nabla f(x^t) \rangle \geq f(e^t) - f(x^t). \tag{20}$$

*Proof.* If $\rho_t = 0$, then $e^t = x^t$ and both inequalities hold with equality. Assume henceforth that $\rho_t > 0$. For (19), we write

$$\langle \nabla f(e^t), \nabla f(x^t) \rangle = \langle \nabla f(e^t) - \nabla f(x^t), \nabla f(x^t) \rangle + \|\nabla f(x^t)\|^2$$

$$\overset{(17)}{=} \frac{1}{\rho_t} \langle \nabla f(e^t) - \nabla f(x^t), e^t - x^t \rangle + \|\nabla f(x^t)\|^2$$

$$\overset{(8)}{\geq} \|\nabla f(x^t)\|^2.$$

For (20), we have

$$\rho_t \langle \nabla f(e^t), \nabla f(x^t) \rangle \overset{(17)}{=} \langle \nabla f(e^t), e^t - x^t \rangle$$

$$\overset{(7)}{\geq} f(e^t) - f(x^t),$$

as wanted. $\square$

**Lemma B.7** (Smooth case). Suppose that $f$ is $L$-smooth. Then the iterates (17)–(18) satisfy

$$\langle \nabla f(e^t), \nabla f(x^t) \rangle \leq (1 + L\rho_t)\|\nabla f(x^t)\|^2, \tag{21}$$

$$\rho_t \langle \nabla f(e^t), \nabla f(x^t) \rangle \leq f(e^t) - f(x^t) + \frac{L\rho_t^2}{2}\|\nabla f(x^t)\|^2, \tag{22}$$

$$(1 - L\rho_t)\|\nabla f(x^t)\| \leq \|\nabla f(e^t)\| \leq (1 + L\rho_t)\|\nabla f(x^t)\|. \tag{23}$$

*Proof.* If $\rho_t = 0$, then $e^t = x^t$ and all statements are immediate. Assume henceforth that $\rho_t > 0$. For (21), we proceed as in the convex case:

$$\langle \nabla f(e^t), \nabla f(x^t) \rangle = \langle \nabla f(e^t) - \nabla f(x^t), \nabla f(x^t) \rangle + \|\nabla f(x^t)\|^2$$

$$\overset{(17)}{=} \frac{1}{\rho_t} \langle \nabla f(e^t) - \nabla f(x^t), e^t - x^t \rangle + \|\nabla f(x^t)\|^2$$

$$\overset{(14)}{\leq} \frac{1}{\rho_t} L\|e^t - x^t\|^2 + \|\nabla f(x^t)\|^2$$

$$= (1 + L\rho_t)\|\nabla f(x^t)\|^2.$$

For (22), apply (10) with $(x, y) = (x^t, e^t)$ to obtain

$$f(x^t) \leq f(e^t) + \langle \nabla f(e^t), x^t - e^t \rangle + \frac{L}{2}\|x^t - e^t\|^2.$$

Rearranging and using (17) gives

$$\langle \nabla f(e^t), e^t - x^t \rangle \leq f(e^t) - f(x^t) + \frac{L}{2}\|e^t - x^t\|^2 = f(e^t) - f(x^t) + \frac{L\rho_t^2}{2}\|\nabla f(x^t)\|^2,$$

which is equivalent to (22). Finally, (23) follows from the triangle inequality and (9):

$$\|\nabla f(e^t)\| \le \|\nabla f(e^t) - \nabla f(x^t)\| + \|\nabla f(x^t)\| \le L\|e^t - x^t\| + \|\nabla f(x^t)\| = (1 + L\rho_t)\|\nabla f(x^t)\|,$$
$$\|\nabla f(e^t)\| \ge \|\nabla f(x^t)\| - \|\nabla f(e^t) - \nabla f(x^t)\| \ge \|\nabla f(x^t)\| - L\|e^t - x^t\| = (1 - L\rho_t)\|\nabla f(x^t)\|.$$

$\square$

Combining Lemmas B.6 and B.7 yields the following collection of bounds that we will invoke frequently.

**Lemma B.8** (Convex and smooth case). Suppose that $f$ is convex and $L$-smooth. Then the iterates (17)–(18) satisfy

$$\|\nabla f(x^t)\|^2 \le \langle \nabla f(e^t), \nabla f(x^t) \rangle \le \|\nabla f(e^t)\|^2, \tag{24}$$

$$\langle \nabla f(e^t), \nabla f(x^t) \rangle \le (1 + L\rho_t)\|\nabla f(x^t)\|^2, \tag{25}$$

$$f(e^t) - f(x^t) \le \rho_t \langle \nabla f(e^t), \nabla f(x^t) \rangle \le f(e^t) - f(x^t) + \frac{L\rho_t^2}{2}\|\nabla f(x^t)\|^2, \tag{26}$$

$$(1 - L\rho_t)\|\nabla f(x^t)\| \le \|\nabla f(e^t)\| \le (1 + L\rho_t)\|\nabla f(x^t)\|. \tag{27}$$

*Proof.* If $\rho_t = 0$, then $e^t = x^t$ and all inequalities hold trivially. Assume henceforth that $\rho_t > 0$. The lower bound in (24) is (19). Applying Cauchy–Schwarz to (19) gives

$$\|\nabla f(x^t)\|^2 \le \langle \nabla f(e^t), \nabla f(x^t) \rangle \le \|\nabla f(e^t)\| \, \|\nabla f(x^t)\|,$$

hence $\|\nabla f(x^t)\| \le \|\nabla f(e^t)\|$. The upper bound in (24) then follows by another application of Cauchy–Schwarz:

$$\langle \nabla f(e^t), \nabla f(x^t) \rangle \le \|\nabla f(e^t)\| \, \|\nabla f(x^t)\| \le \|\nabla f(e^t)\|^2.$$

The remaining statements are direct consequences of Lemmas B.6 and B.7: (25) follows from (21), (26) from (20) and (22), and (27) from (23). $\square$

## C. Proofs for Section 2: Non-negativity and descent property of Polyak Scheduler

This section provides the proofs of the basic properties established in Section 2. For completeness, we restate the two propositions proved below.

**Proposition C.1** (Non-negativity and lower bound for Polyak Scheduler). *Let $f$ be convex and $L$-smooth. If $\rho_t \leq 1/L$, then*

$$\gamma_t \geq \frac{1 - L\rho_t}{2L(1 + L\rho_t)} \geq 0.$$

*In particular, in this regime the ReLU safeguard is redundant and the Polyak step size for the deterministic USAM can be written as*

$$\gamma_t = \frac{f(e^t) - f^* - \rho_t \langle \nabla f(e^t), \nabla f(x^t) \rangle}{\|\nabla f(e^t)\|^2}.$$

*Proof of Proposition 2.1.* We first show that the numerator in Polyak Scheduler is non-negative when $\rho_t \leq 1/L$. We have

$$
f(e^t) - f^* - \rho_t \langle \nabla f(e^t), \nabla f(x^t) \rangle \overset{(26)}{\geq} f(e^t) - f^* - \left( f(e^t) - f(x^t) + \frac{L\rho_t^2}{2} \|\nabla f(x^t)\|^2 \right)
$$

$$
= f(x^t) - f^* - \frac{L\rho_t^2}{2} \|\nabla f(x^t)\|^2
$$

$$
\overset{(15)}{\geq} \frac{1}{2L} \|\nabla f(x^t)\|^2 - \frac{L\rho_t^2}{2} \|\nabla f(x^t)\|^2
$$

$$
= \frac{1 - L^2\rho_t^2}{2L} \|\nabla f(x^t)\|^2 \geq 0. \tag{28}
$$

Hence, in this regime,

$$\left[ f(e^t) - f^* - \rho_t \langle \nabla f(e^t), \nabla f(x^t) \rangle \right]_+ = f(e^t) - f^* - \rho_t \langle \nabla f(e^t), \nabla f(x^t) \rangle,$$

so the ReLU can be dropped. Next, assume $\nabla f(x^t) \neq 0$ (otherwise $x^t \in X^*$ and the algorithm terminates). Then, by definition,

$$
\gamma_t = \frac{f(e^t) - f^* - \rho_t \langle \nabla f(e^t), \nabla f(x^t) \rangle}{\|\nabla f(e^t)\|^2}
$$

$$
\overset{(28)}{\geq} \frac{\frac{1 - L^2\rho_t^2}{2L} \|\nabla f(x^t)\|^2}{\|\nabla f(e^t)\|^2}
$$

$$
\overset{(27)}{\geq} \frac{\frac{1 - L^2\rho_t^2}{2L} \|\nabla f(x^t)\|^2}{(1 + L\rho_t)^2 \|\nabla f(x^t)\|^2}
$$

$$
= \frac{1 - L^2\rho_t^2}{2L(1 + L\rho_t)^2}
$$

$$
= \frac{1 - L\rho_t}{2L(1 + L\rho_t)}.
$$

This concludes the proof. $\qquad\square$

**Proposition C.2** (Descent property of Polyak Scheduler). *Let $f$ be convex and $L$-smooth, and suppose $\rho_t \leq 1/L$. Then the iterates generated by deterministic USAM with Polyak Scheduler satisfy, for all $t \geq 0$,*

$$\|x^{t+1} - x^*\|^2 \leq \|x^t - x^*\|^2 - \frac{(1 - L\rho_t)^2}{2L} \left( f(x^t) - f^* \right).$$

In particular, the sequence $\{\|x^t - x^*\|\}_{t \geq 0}$ is non-increasing.

*Proof of Proposition 2.2.* Assume $\nabla f(x^t) \neq 0$ (otherwise $x^t \in X^*$ and the algorithm terminates). Starting from the USAM update (18) and expanding the squared distance gives

$$
\begin{aligned}
\|x^{t+1} - x^*\|^2 &= \|x^t - x^*\|^2 - 2\gamma_t \langle \nabla f(e^t), x^t - x^* \rangle + \gamma_t^2 \|\nabla f(e^t)\|^2 \\
&= \|x^t - x^*\|^2 - 2\gamma_t \left( \langle \nabla f(e^t), e^t - x^* \rangle - \langle \nabla f(e^t), e^t - x^t \rangle \right) + \gamma_t^2 \|\nabla f(e^t)\|^2 \\
&\overset{(7)}{\leq} \|x^t - x^*\|^2 - 2\gamma_t \left( f(e^t) - f^* - \langle \nabla f(e^t), e^t - x^t \rangle \right) + \gamma_t^2 \|\nabla f(e^t)\|^2,
\end{aligned}
$$

With the choice of step size (Polyak Scheduler), the last expression becomes

$$
\|x^{t+1} - x^*\|^2 \leq \|x^t - x^*\|^2 - \frac{\left( f(e^t) - f^* - \langle \nabla f(e^t), e^t - x^t \rangle \right)^2}{\|\nabla f(e^t)\|^2}.
$$

We now bound the quantity $f(e^t) - f^* - \langle \nabla f(e^t), e^t - x^t \rangle = f(e^t) - f^* - \rho_t \langle \nabla f(e^t), \nabla f(x^t) \rangle$ from below in terms of function values. From (26), $\rho_t \langle \nabla f(e^t), \nabla f(x^t) \rangle \leq f(e^t) - f(x^t) + \frac{L\rho_t^2}{2} \|\nabla f(x^t)\|^2$, so

$$
f(e^t) - f^* - \langle \nabla f(e^t), e^t - x^t \rangle \geq f(x^t) - f^* - \frac{L\rho_t^2}{2} \|\nabla f(x^t)\|^2.
$$

By smoothness (15), $\|\nabla f(x^t)\|^2 \leq 2L(f(x^t) - f^*)$, hence

$$
f(e^t) - f^* - \langle \nabla f(e^t), e^t - x^t \rangle \geq (1 - L^2\rho_t^2)(f(x^t) - f^*) = (1 - L\rho_t)(1 + L\rho_t)(f(x^t) - f^*). \tag{29}
$$

Next, we bound $\|\nabla f(e^t)\|^2$ from above. By (27), $\|\nabla f(e^t)\|^2 \leq (1 + L\rho_t)^2 \|\nabla f(x^t)\|^2 \leq 2L(1 + L\rho_t)^2 (f(x^t) - f^*)$, where the last step uses (15). Combining these bounds:

$$
\frac{\left( f(e^t) - f^* - \langle \nabla f(e^t), e^t - x^t \rangle \right)^2}{\|\nabla f(e^t)\|^2} \geq \frac{(1 - L\rho_t)^2(1 + L\rho_t)^2 (f(x^t) - f^*)^2}{2L(1 + L\rho_t)^2(f(x^t) - f^*)} = \frac{(1 - L\rho_t)^2}{2L}\left( f(x^t) - f^* \right).
$$

Hence, we obtain the desired inequality

$$
\|x^{t+1} - x^*\|^2 \leq \|x^t - x^*\|^2 - \frac{(1 - L\rho_t)^2}{2L}\left( f(x^t) - f^* \right).
$$

$\qquad\square$

# D. Proofs for Section 3: Convergence Guarantees

## D.1. Deterministic

We begin with the deterministic (full-batch) setting and prove the convergence guarantees stated in Section 3. Throughout this subsection we consider the USAM (17)–(18) equipped with the Polyak step size (Polyak Scheduler), and we assume a constant sharpness radius $\rho_t = \rho$.

**Theorem D.1** (Strongly convex case). Let $f$ be $\mu$-strongly convex and $L$-smooth. Suppose that $\rho_t = \rho \leq \frac{1}{L}$. Then the iterates generated by the deterministic USAM (17)–(18) with step size (Polyak Scheduler) satisfy, for all $t \geq 0$,

$$\|x^t - x^*\|^2 \leq \left(1 - \frac{\mu(1 - L\rho)^2}{4L}\right)^t \|x^0 - x^*\|^2.$$

*Proof of Theorem 3.1.* Starting from (4), we have for each $t \geq 0$,

$$\|x^{t+1} - x^*\|^2 \leq \|x^t - x^*\|^2 - \frac{(1 - L\rho)^2}{2L} \left(f(x^t) - f^*\right).$$

Since $f$ is $\mu$-strongly convex, the quadratic growth inequality (12) implies

$$f(x^t) - f^* \geq \frac{\mu}{2}\|x^t - x^*\|^2.$$

Substituting this into the descent inequality yields the contraction

$$\begin{aligned}
\|x^{t+1} - x^*\|^2 &\leq \|x^t - x^*\|^2 - \frac{(1 - L\rho)^2}{2L} \cdot \frac{\mu}{2} \|x^t - x^*\|^2 \\
&= \left(1 - \frac{\mu(1 - L\rho)^2}{4L}\right) \|x^t - x^*\|^2.
\end{aligned}$$

Unrolling the recurrence gives the stated linear convergence rate. □

**Theorem D.2** (Convex case). Let $f$ be convex and $L$-smooth. Suppose that $\rho_t = \rho \leq \frac{1}{L}$. Then the iterates generated by the deterministic USAM (17)–(18) with step size (Polyak Scheduler) satisfy, for all $T \geq 1$,

$$f(\overline{x}^T) - f^* \leq \frac{2L\|x^0 - x^*\|^2}{T(1 - L\rho)^2},$$

where $\overline{x}^T = \frac{1}{T}\sum_{t=0}^{T-1} x^t$ is the Cesaro average.

*Proof of Theorem 3.2.* By the descent inequality (4), for each $t \geq 0$ we have

$$\|x^{t+1} - x^*\|^2 \leq \|x^t - x^*\|^2 - \frac{(1 - L\rho)^2}{2L} \left(f(x^t) - f^*\right).$$

Summing this inequality over $t = 0, 1, \ldots, T - 1$ yields the telescoping bound

$$\frac{(1 - L\rho)^2}{2L} \sum_{t=0}^{T-1} \left(f(x^t) - f^*\right) \leq \|x^0 - x^*\|^2 - \|x^T - x^*\|^2 \leq \|x^0 - x^*\|^2.$$

Dividing both sides by $T$ and using Jensen's inequality gives

$$\begin{aligned}
f(\overline{x}^T) - f^* &\leq \frac{1}{T} \sum_{t=0}^{T-1} \left(f(x^t) - f^*\right) \\
&\leq \frac{2L\|x^0 - x^*\|^2}{(1 - L\rho)^2},
\end{aligned}$$

which is the desired result. □

We finally consider the deterministic setting with a decreasing sharpness radius. The main point is that once $\rho_t$ becomes sufficiently small (which holds eventually by assumption), the descent inequality (4) yields summability of the squared gradient norms.

**Theorem D.3** (Decreasing radius implies vanishing gradients). Let $f$ be convex and $L$-smooth. Consider the deterministic USAM iterates (17)–(18) with step size (Polyak Scheduler), where the radii $(\rho_t)_{t \geq 0}$ satisfy $\rho_t \downarrow 0$, meaning that $\rho_t \geq 0$, $\rho_{t+1} \leq \rho_t$ for all $t$, and $\rho_t \to 0$. Then

$$\sum_{t=0}^{\infty} \left( f(x^t) - f^* \right) < \infty,$$

and consequently $f(x^t) \to f^*$ as $t \to \infty$.

*Proof of Theorem 3.4.* Since $\rho_t \downarrow 0$, there exists an index $t_0 \in \mathbb{N}$ such that $\rho_{t_0} < \frac{1}{2L}$. By monotonicity of $(\rho_t)$, we then have $\rho_t \leq \frac{1}{2L}$ for all $t \geq t_0$, which implies $1 - L\rho_t \geq \frac{1}{2}$ and hence

$$\frac{(1 - L\rho_t)^2}{2L} \geq \frac{1}{8L} \qquad \forall t \geq t_0.$$

For all $t \geq t_0$, we may therefore apply (4) to obtain

$$\|x^{t+1} - x^*\|^2 \leq \|x^t - x^*\|^2 - \frac{(1 - L\rho_t)^2}{2L} \left( f(x^t) - f^* \right)$$

$$\leq \|x^t - x^*\|^2 - \frac{1}{8L} \left( f(x^t) - f^* \right).$$

Summing (telescoping) from $t = t_0$ to $T - 1$ for any $T \geq t_0 + 1$ yields

$$\frac{1}{8L} \sum_{t=t_0}^{T-1} \left( f(x^t) - f^* \right) \leq \|x^{t_0} - x^*\|^2 - \|x^T - x^*\|^2 \leq \|x^{t_0} - x^*\|^2.$$

Letting $T \to \infty$ gives $\sum_{t=t_0}^{\infty} \left( f(x^t) - f^* \right) < \infty$. Since the prefix sum $\sum_{t=0}^{t_0 - 1} \|\nabla f(x^t)\|^2$ is finite, we conclude

$$\sum_{t=0}^{\infty} \left( f(x^t) - f^* \right) < \infty.$$

Finally, because the summands are non-negative, summability implies $(f(x^t) - f^*) \to 0$, i.e., $f(x^t) \to f^*$. $\qquad\square$

### D.2. Stochastic

In this subsection, we focus on the proofs of the convergence guarantees in the stochastic setting. Recall that each component function $f_i$ is convex and $L_i$-smooth. We let $L_{\max} = \max_{i \in [n]} L_i$, so each $f_i$ is also $L_{\max}$-smooth. We analyze USAM equipped with the capped Stochastic Polyak Scheduler. Throughout, we assume a constant radius $\rho_t = \rho$ and $\rho \leq 1/L_{\max}$.

We begin by establishing a key descent inequality that underlies the proofs of both Theorems 3.5 and 3.8.

**Proposition D.4** (Stochastic descent inequality). Let each $f_i$ be convex and $L_i$-smooth, and set $L_{\max} = \max_{i \in [n]} L_i$. Suppose that $\rho_t = \rho \leq \frac{1}{L_{\max}}$. Let $\alpha = (1 - L_{\max}\rho)^2 \min \left\{ \frac{1}{2L_{\max}}, \gamma_b \right\}$. Then the iterates of USAM with Stochastic Polyak Scheduler satisfy, for all $t \geq 0$,

$$\mathbb{E}\|x^{t+1} - x^*\|^2 - \mathbb{E}\|x^t - x^*\|^2 \leq -\alpha \, \mathbb{E}[f(x^t) - f^*] + (2\gamma_b - \alpha)\sigma^2. \tag{30}$$

*Proof.* Fix $t \geq 0$. Using the update $x^{t+1} = x^t - \gamma_t \nabla f_{S_t}(e^t)$ and expanding the squared distance gives

$$\|x^{t+1} - x^*\|^2 - \|x^t - x^*\|^2 = -2\gamma_t \langle \nabla f_{S_t}(e^t), x^t - x^* \rangle + \gamma_t^2 \|\nabla f_{S_t}(e^t)\|^2$$

$$= -2\gamma_t \left( \langle \nabla f_{S_t}(e^t), e^t - x^* \rangle - \langle \nabla f_{S_t}(e^t), e^t - x^t \rangle \right) + \gamma_t^2 \|\nabla f_{S_t}(e^t)\|^2$$

$$\overset{(7)}{\leq} -2\gamma_t \left( f_{S_t}(e^t) - f_{S_t}(x^*) - \langle \nabla f_{S_t}(e^t), e^t - x^t \rangle \right) + \gamma_t^2 \|\nabla f_{S_t}(e^t)\|^2. \tag{31}$$

Introduce the shorthand

$$A_t = f_{S_t}(e^t) - \ell_{S_t}^* - \langle \nabla f_{S_t}(e^t), e^t - x^t \rangle = f_{S_t}(e^t) - \ell_{S_t}^* - \rho \langle \nabla f_{S_t}(e^t), \nabla f_{S_t}(x^t) \rangle$$

so that

$$f_{S_t}(e^t) - f_{S_t}(x^*) - \langle \nabla f_{S_t}(e^t), e^t - x^t \rangle = A_t + \ell_{S_t}^* - f_{S_t}(x^*).$$

Substituting this identity into (31) yields

$$\|x^{t+1} - x^*\|^2 - \|x^t - x^*\|^2 \leq -2\gamma_t A_t + \gamma_t^2 \|\nabla f_{S_t}(e^t)\|^2 + 2\gamma_t \left( f_{S_t}(x^*) - \ell_{S_t}^* \right). \tag{32}$$

Since each $f_i$ is $L_{\max}$-smooth, so is $f_{S_t}$, and because $\rho \leq 1/L_{\max}$, applying the improved lower bound (as in the proof of Proposition 2.2) to the mini-batch function $f_{S_t}$ gives

$$A_t \geq (1 - L_{\max}^2 \rho^2)(f_{S_t}(x^t) - \ell_{S_t}^*) \geq (1 - L_{\max}\rho)^2 (f_{S_t}(x^t) - \ell_{S_t}^*), \tag{33}$$

where the last step uses $(1 - L_{\max}^2 \rho^2) = (1 - L_{\max}\rho)(1 + L_{\max}\rho) \geq (1 - L_{\max}\rho)^2$. We now bound the right-hand side of (32) using the two cases in the definition of $\gamma_t$.

*Case 1:* $\gamma_t = \frac{A_t}{\|\nabla f_{S_t}(e^t)\|^2} \leq \gamma_b$. Then the first two terms in (32) combine as

$$-2\gamma_t A_t + \gamma_t^2 \|\nabla f_{S_t}(e^t)\|^2 = -\frac{A_t^2}{\|\nabla f_{S_t}(e^t)\|^2},$$

and hence

$$\|x^{t+1} - x^*\|^2 - \|x^t - x^*\|^2 \leq -\frac{A_t^2}{\|\nabla f_{S_t}(e^t)\|^2} + 2\gamma_t \left( f_{S_t}(x^*) - \ell_{S_t}^* \right)$$

$$\leq -\frac{A_t^2}{\|\nabla f_{S_t}(e^t)\|^2} + 2\gamma_b \left( f_{S_t}(x^*) - \ell_{S_t}^* \right). \tag{34}$$

Moreover, by (27) and (15) applied to $f_{S_t}$, we have $\|\nabla f_{S_t}(e^t)\|^2 \leq 2L_{\max}(1 + L_{\max}\rho)^2 (f_{S_t}(x^t) - \ell_{S_t}^*)$. Therefore

$$\frac{A_t^2}{\|\nabla f_{S_t}(e^t)\|^2} \geq \frac{(1 - L_{\max}\rho)^4 (f_{S_t}(x^t) - \ell_{S_t}^*)^2}{2L_{\max}(1 + L_{\max}\rho)^2 (f_{S_t}(x^t) - \ell_{S_t}^*)}$$

$$= \frac{(1 - L_{\max}\rho)^4}{2L_{\max}(1 + L_{\max}\rho)^2} \left( f_{S_t}(x^t) - \ell_{S_t}^* \right)$$

$$\geq \frac{(1 - L_{\max}\rho)^2}{2L_{\max}} \left( f_{S_t}(x^t) - \ell_{S_t}^* \right), \tag{35}$$

where the last inequality uses $\frac{(1 - L_{\max}\rho)^2}{(1 + L_{\max}\rho)^2} \leq 1$. Combining (34) and (35) yields

$$\|x^{t+1} - x^*\|^2 - \|x^t - x^*\|^2 \leq -\frac{(1 - L_{\max}\rho)^2}{2L_{\max}} (f_{S_t}(x^t) - \ell_{S_t}^*) + 2\gamma_b \left( f_{S_t}(x^*) - \ell_{S_t}^* \right). \tag{36}$$

*Case 2:* $\gamma_t = \gamma_b$, which means $\frac{A_t}{\|\nabla f_{S_t}(e^t)\|^2} \geq \gamma_b$, i.e., $A_t \geq \gamma_b \|\nabla f_{S_t}(e^t)\|^2$. Plugging $\gamma_t = \gamma_b$ into (32) gives

$$\|x^{t+1} - x^*\|^2 - \|x^t - x^*\|^2 \leq -2\gamma_b A_t + \gamma_b^2 \|\nabla f_{S_t}(e^t)\|^2 + 2\gamma_b \left( f_{S_t}(x^*) - \ell_{S_t}^* \right)$$

$$\leq -\gamma_b A_t + 2\gamma_b \left( f_{S_t}(x^*) - \ell_{S_t}^* \right),$$

where we used $A_t \geq \gamma_b \|\nabla f_{S_t}(e^t)\|^2$ in the last step. Using (33), we obtain

$$\|x^{t+1} - x^*\|^2 - \|x^t - x^*\|^2 \leq -\gamma_b(1 - L_{\max}\rho)^2(f_{S_t}(x^t) - \ell_{S_t}^*) + 2\gamma_b\left(f_{S_t}(x^*) - \ell_{S_t}^*\right). \tag{37}$$

Combining (36) and (37), and recalling that $\alpha = (1 - L_{\max}\rho)^2 \min\{\frac{1}{2L_{\max}}, \gamma_b\}$, we obtain in all cases

$$\|x^{t+1} - x^*\|^2 - \|x^t - x^*\|^2 \leq -\alpha\left(f_{S_t}(x^t) - \ell_{S_t}^*\right) + 2\gamma_b\left(f_{S_t}(x^*) - \ell_{S_t}^*\right). \tag{38}$$

Now take conditional expectation on $x^t$. Note that

$$\mathbb{E}_t[f_{S_t}(x^t) - \ell_{S_t}^*] = f(x^t) - \mathbb{E}[\ell_{S_t}^*] = (f(x^t) - f^*) + \sigma^2,$$
$$\mathbb{E}_t[f_{S_t}(x^*) - \ell_{S_t}^*] = \sigma^2.$$

Taking conditional expectations in (38) and substituting:

$$\mathbb{E}_t\|x^{t+1} - x^*\|^2 - \|x^t - x^*\|^2 \leq -\alpha\left(f(x^t) - f^*\right) + (2\gamma_b - \alpha)\sigma^2. \tag{39}$$

Taking full expectations in (39) and using the tower property yields (30). $\qquad\square$

We now use (30) to prove the two main stochastic convergence guarantees.

**Theorem D.5** (Strongly convex case). Let each $f_i$ be convex and $L_i$-smooth, and set $L_{\max} = \max_{i \in [n]} L_i$. Suppose that $f$ is $\mu$-strongly convex and that $\rho_t = \rho \leq \frac{1}{L_{\max}}$. Let $\alpha = (1 - L_{\max}\rho)^2 \min\left\{\frac{1}{2L_{\max}}, \gamma_b\right\}$. Then the iterates of USAM with Stochastic Polyak Scheduler satisfy, for all $t \geq 0$,

$$\mathbb{E}\|x^t - x^*\|^2 \leq \left(1 - \frac{\mu\alpha}{2}\right)^t \|x^0 - x^*\|^2 + \frac{2(2\gamma_b - \alpha)}{\mu\alpha}\sigma^2,$$

where $\sigma^2 = \mathbb{E}\left[f_{S_t}(x^*) - \ell_{S_t}^*\right]$.

*Proof of Theorem 3.5.* Starting from (30), we have for all $t \geq 0$,

$$\mathbb{E}\|x^{t+1} - x^*\|^2 \leq \mathbb{E}\|x^t - x^*\|^2 - \alpha\,\mathbb{E}[f(x^t) - f^*] + (2\gamma_b - \alpha)\sigma^2.$$

Since $f$ is $\mu$-strongly convex, the quadratic growth inequality (12) implies $f(x^t) - f^* \geq \frac{\mu}{2}\|x^t - x^*\|^2$, and therefore

$$\mathbb{E}\|x^{t+1} - x^*\|^2 \leq \left(1 - \frac{\mu\alpha}{2}\right)\mathbb{E}\|x^t - x^*\|^2 + (2\gamma_b - \alpha)\sigma^2.$$

Unrolling this linear recursion yields

$$\mathbb{E}\|x^t - x^*\|^2 \leq \left(1 - \frac{\mu\alpha}{2}\right)^t \|x^0 - x^*\|^2 + \frac{2(2\gamma_b - \alpha)}{\mu\alpha}\sigma^2,$$

which is the desired result. $\qquad\square$

**Theorem D.6** (Convex case). Let each $f_i$ be convex and $L_i$-smooth, and set $L_{\max} = \max_{i \in [n]} L_i$. Suppose that $\rho_t = \rho \leq \frac{1}{L_{\max}}$. Let $\alpha = (1 - L_{\max}\rho)^2 \min\left\{\frac{1}{2L_{\max}}, \gamma_b\right\}$. Then the iterates of USAM with Stochastic Polyak Scheduler satisfy, for all $T \geq 1$,

$$\frac{1}{T}\sum_{t=0}^{T-1} \mathbb{E}[f(x^t) - f^*] \leq \frac{\|x^0 - x^*\|^2}{\alpha T} + \frac{2\gamma_b - \alpha}{\alpha}\sigma^2,$$

where $\sigma^2 = \mathbb{E}\left[f_{S_t}(x^*) - \ell_{S_t}^*\right]$.

*Proof of Theorem 3.8.* Summing (30) for $t = 0, \ldots, T-1$ and telescoping gives

$$\alpha\sum_{t=0}^{T-1} \mathbb{E}[f(x^t) - f^*] \leq \|x^0 - x^*\|^2 + (2\gamma_b - \alpha)\sigma^2 T.$$

Dividing by $\alpha T$ and applying Jensen's inequality concludes the proof. $\qquad\square$

# E. Further Numerical Experiments

This section collects additional implementation details and experimental results for the deep-learning benchmarks. In particular, we provide (i) the training protocol and the hyper-parameter ranges used across all CIFAR experiments (Table 5), (ii) pseudocode for the Polyak Scheduler variant of USAM used in our runs (Algorithm 1), and (iii) additional CIFAR-10/100 test accuracies for ResNet-20/32 under different weight-decay settings and sharpness radii (see Section E.3).

## E.1. Deep-learning protocol and hyper-parameters

All CIFAR experiments are run for 100 epochs with batch size 128. We evaluate top-1 test accuracy and report mean $\pm$ standard deviation over 3 random seeds. We consider two weight-decay settings, $wd \in \{0, 5 \cdot 10^{-4}\}$, and a range of sharpness radii $\rho \in \{0.1, 0.2, 0.3, 0.4\}$. For the constant and cosine-annealing schedulers we tune the initial learning rate over $\gamma \in \{10^{-3}, 10^{-2}, 10^{-1}\}$. A summary of these choices is given in Table 5.

| Hyper-parameter | Value |
|---|---|
| Datasets | CIFAR-10/100 (Krizhevsky et al., 2009) |
| Architectures | ResNet 20/32 (He et al., 2016) |
| Hardware | NVIDIA RTX 6000 Ada Generation |
| Epochs | 100 |
| Batch-size | 128 |
| Weight Decay | $0.0, 0.0005$ |
| Momentum | $0.0$ |
| Optimizers | USAM, SAM |
| Sharpness Radius ($\rho$) | $0.1, 0.2, 0.3, 0.4$ |
| Constant / Cosine LR | $\gamma \in \{10^{-3}, 10^{-2}, 10^{-1}\}$ |
| Schedulers | Constant, Cosine Annealing, Stochastic Polyak Scheduler |
| SPS cap | $\gamma_b \in \{10^{-3}, 10^{-2}, 10^{-1}, 10^0, 10^1, 10^2, 10^3\}$ |
| Reporting | Mean $\pm$ std over 3 seeds |

*Table 5.* Experimental details

## E.2. Pseudocode

For completeness, we provide pseudocode for USAM with Stochastic Polyak Scheduler used in our experiments.

---

**Algorithm 1** USAM with Stochastic Polyak Scheduler

---

**Require:** $x^0 \in \mathbb{R}^d$, iterations $T$, radius $\rho$, mini-batch size $\tau$, lower bounds $\ell_{S_t}^*$, cap $\gamma_b > 0$
1: **for** $t = 0$ **to** $T - 1$ **do**
2:     Sample a mini-batch $S_t \subseteq [n]$ with $|S_t| = \tau$
3:     $e^t \leftarrow x^t + \rho_t \nabla f_{S_t}(x^t)$
4:     $\gamma_t \leftarrow \min \left\{ \dfrac{f_{S_t}(e^t) - \ell_{S_t}^* - \rho_t \langle \nabla f_{S_t}(e^t), e^t - x^t \rangle}{\|\nabla f_{S_t}(e^t)\|^2}, \; \gamma_b \right\}$
5:     $x^{t+1} \leftarrow x^t - \gamma_t \nabla f_{S_t}(e^t)$
6: **end for**

---

## E.3. Additional CIFAR results

We now report additional test accuracies for both USAM and SAM. Unless stated otherwise, all entries are mean $\pm$ standard deviation over 3 seeds, and **Best** is computed row-wise.

*Table 6.* CIFAR-10 test accuracy (%, mean $\pm$ std over 3 seeds) for ResNet-20 trained with USAM under three learning-rate schedules with $wd = 0.0$. Best in bold.

| $wd = 0.0$ | Constant USAM | USAM with Cosine Annealing | USAM with Stochastic Polyak Scheduler |
| --- | --- | --- | --- |
| $\rho = 0.1$ | $88.41_{\pm 0.11}$ | $87.50_{\pm 0.07}$ | $\mathbf{88.96_{\pm 0.12}}$ |
| $\rho = 0.2$ | $88.13_{\pm 0.14}$ | $86.19_{\pm 0.26}$ | $\mathbf{89.40_{\pm 0.16}}$ |
| $\rho = 0.3$ | $87.93_{\pm 0.06}$ | $85.04_{\pm 0.24}$ | $\mathbf{89.69_{\pm 0.11}}$ |
| $\rho = 0.4$ | $87.65_{\pm 0.11}$ | $84.32_{\pm 0.20}$ | $\mathbf{89.64_{\pm 0.19}}$ |

*Table 7.* CIFAR-10 test accuracy (%, mean $\pm$ std over 3 seeds) for ResNet-20 trained with USAM under three learning-rate schedules with $wd = 5 \cdot 10^{-4}$. Best in bold.

| $wd = 5 \cdot 10^{-4}$ | Constant USAM | USAM with Cosine Annealing | USAM with Stochastic Polyak Scheduler |
| --- | --- | --- | --- |
| $\rho = 0.1$ | $89.34_{\pm 0.23}$ | $89.25_{\pm 0.28}$ | $\mathbf{91.16_{\pm 0.30}}$ |
| $\rho = 0.2$ | $89.58_{\pm 0.12}$ | $88.00_{\pm 0.15}$ | $\mathbf{91.18_{\pm 0.04}}$ |
| $\rho = 0.3$ | $89.18_{\pm 0.12}$ | $87.02_{\pm 0.07}$ | $\mathbf{90.65_{\pm 0.20}}$ |
| $\rho = 0.4$ | $88.55_{\pm 0.18}$ | $85.81_{\pm 0.41}$ | $\mathbf{90.30_{\pm 0.20}}$ |

*Table 8.* CIFAR-100 test accuracy (%, mean $\pm$ std over 3 seeds) for ResNet-20 trained with USAM under three learning-rate schedules with $wd = 0.0$. Best in bold.

| $wd = 0.0$ | Constant USAM | USAM with Cosine Annealing | USAM with Stochastic Polyak Scheduler |
| --- | --- | --- | --- |
| $\rho = 0.1$ | $88.42_{\pm 0.23}$ | $87.61_{\pm 0.33}$ | $\mathbf{89.32_{\pm 0.32}}$ |
| $\rho = 0.2$ | $88.45_{\pm 0.01}$ | $86.62_{\pm 0.02}$ | $\mathbf{89.36_{\pm 0.12}}$ |
| $\rho = 0.3$ | $88.23_{\pm 0.02}$ | $85.66_{\pm 0.16}$ | $\mathbf{89.67_{\pm 0.04}}$ |
| $\rho = 0.4$ | $87.47_{\pm 0.29}$ | $84.10_{\pm 0.39}$ | $\mathbf{89.67_{\pm 0.15}}$ |

*Table 9.* CIFAR-100 test accuracy (%, mean $\pm$ std over 3 seeds) for ResNet-20 trained with USAM under three learning-rate schedules with $wd = 5 \cdot 10^{-4}$. Best in bold.

| $wd = 5 \cdot 10^{-4}$ | Constant USAM | USAM with Cosine Annealing | USAM with Stochastic Polyak Scheduler |
| --- | --- | --- | --- |
| $\rho = 0.1$ | $89.63_{\pm 0.24}$ | $89.44_{\pm 0.17}$ | $\mathbf{90.94_{\pm 0.12}}$ |
| $\rho = 0.2$ | $89.51_{\pm 0.03}$ | $88.16_{\pm 0.08}$ | $\mathbf{91.17_{\pm 0.23}}$ |
| $\rho = 0.3$ | $89.18_{\pm 0.11}$ | $86.96_{\pm 0.27}$ | $\mathbf{90.82_{\pm 0.05}}$ |
| $\rho = 0.4$ | $88.62_{\pm 0.13}$ | $86.13_{\pm 0.14}$ | $\mathbf{90.42_{\pm 0.06}}$ |

*Table 10.* CIFAR-10 test accuracy (%, mean $\pm$ std over 3 seeds) for ResNet-20 trained with SAM under three learning-rate schedules with $wd = 0.0$. Best in bold.

| $wd = 0.0$ | Constant SAM | SAM with Cosine Annealing | SAM with Stochastic Polyak Scheduler |
| --- | --- | --- | --- |
| $\rho = 0.1$ | $88.53_{\pm 0.14}$ | $87.63_{\pm 0.09}$ | $\mathbf{89.34_{\pm 0.07}}$ |
| $\rho = 0.2$ | $88.31_{\pm 0.20}$ | $86.21_{\pm 0.47}$ | $\mathbf{89.93_{\pm 0.09}}$ |
| $\rho = 0.3$ | $87.17_{\pm 0.11}$ | $83.66_{\pm 0.19}$ | $\mathbf{90.11_{\pm 0.18}}$ |
| $\rho = 0.4$ | $85.57_{\pm 0.50}$ | $80.63_{\pm 0.27}$ | $\mathbf{89.81_{\pm 0.21}}$ |

*Table 11.* CIFAR-10 test accuracy (%, mean $\pm$ std over 3 seeds) for ResNet-20 trained with SAM under three learning-rate schedules with $wd = 5 \cdot 10^{-4}$. Best in bold.

| $wd = 5 \cdot 10^{-4}$ | Constant SAM | SAM with Cosine Annealing | SAM with Stochastic Polyak Scheduler |
| --- | --- | --- | --- |
| $\rho = 0.1$ | $89.58_{\pm 0.23}$ | $89.48_{\pm 0.04}$ | $\mathbf{90.94_{\pm 0.07}}$ |
| $\rho = 0.2$ | $89.40_{\pm 0.31}$ | $88.16_{\pm 0.21}$ | $\mathbf{91.17_{\pm 0.08}}$ |
| $\rho = 0.3$ | $88.68_{\pm 0.26}$ | $85.94_{\pm 0.02}$ | $\mathbf{90.22_{\pm 0.26}}$ |
| $\rho = 0.4$ | $87.44_{\pm 0.23}$ | $83.65_{\pm 0.24}$ | $\mathbf{89.25_{\pm 0.17}}$ |

*Table 12.* CIFAR-100 test accuracy (%, mean $\pm$ std over 3 seeds) for ResNet-20 trained with SAM under three learning-rate schedules with $wd = 0.0$. Best in bold.

| $wd = 0.0$ | Constant SAM | SAM with Cosine Annealing | SAM with Stochastic Polyak Scheduler |
|---|---|---|---|
| $\rho = 0.1$ | $88.58_{\pm 0.08}$ | $87.88_{\pm 0.09}$ | $\mathbf{89.32_{\pm 0.34}}$ |
| $\rho = 0.2$ | $88.54_{\pm 0.09}$ | $86.22_{\pm 0.24}$ | $\mathbf{89.98_{\pm 0.16}}$ |
| $\rho = 0.3$ | $87.49_{\pm 0.35}$ | $84.28_{\pm 0.09}$ | $\mathbf{90.03_{\pm 0.14}}$ |
| $\rho = 0.4$ | $86.05_{\pm 0.21}$ | $81.69_{\pm 0.42}$ | $\mathbf{90.33_{\pm 0.12}}$ |

*Table 13.* CIFAR-100 test accuracy (%, mean $\pm$ std over 3 seeds) for ResNet-20 trained with SAM under three learning-rate schedules with $wd = 5 \cdot 10^{-4}$. Best in bold.

| $wd = 5 \cdot 10^{-4}$ | Constant SAM | SAM with Cosine Annealing | SAM with Stochastic Polyak Scheduler |
|---|---|---|---|
| $\rho = 0.1$ | $89.56_{\pm 0.21}$ | $89.80_{\pm 0.13}$ | $\mathbf{91.15_{\pm 0.02}}$ |
| $\rho = 0.2$ | $89.53_{\pm 0.25}$ | $88.13_{\pm 0.02}$ | $\mathbf{91.28_{\pm 0.09}}$ |
| $\rho = 0.3$ | $88.92_{\pm 0.13}$ | $86.05_{\pm 0.37}$ | $\mathbf{90.39_{\pm 0.09}}$ |
| $\rho = 0.4$ | $87.85_{\pm 0.18}$ | $84.31_{\pm 0.27}$ | $\mathbf{89.29_{\pm 0.08}}$ |

*Table 14.* CIFAR-10 test accuracy (%, mean $\pm$ std over 3 seeds) for ResNet-32 trained with USAM under three learning-rate schedules with $wd = 0.0$. Best in bold.

| $wd = 0.0$ | Constant USAM | USAM with Cosine Annealing | USAM with Stochastic Polyak Scheduler |
|---|---|---|---|
| $\rho = 0.1$ | $88.93_{\pm 0.14}$ | $87.88_{\pm 0.17}$ | $\mathbf{89.43_{\pm 0.15}}$ |
| $\rho = 0.2$ | $88.97_{\pm 0.14}$ | $87.06_{\pm 0.19}$ | $\mathbf{89.72_{\pm 0.20}}$ |
| $\rho = 0.3$ | $88.69_{\pm 0.21}$ | $85.69_{\pm 0.24}$ | $\mathbf{90.19_{\pm 0.05}}$ |
| $\rho = 0.4$ | $88.24_{\pm 0.06}$ | $84.77_{\pm 0.19}$ | $\mathbf{90.26_{\pm 0.04}}$ |

*Table 15.* CIFAR-10 test accuracy (%, mean $\pm$ std over 3 seeds) for ResNet-32 trained with USAM under three learning-rate schedules with $wd = 5 \cdot 10^{-4}$. Best in bold.

| $wd = 5 \cdot 10^{-4}$ | Constant USAM | USAM with Cosine Annealing | USAM with Stochastic Polyak Scheduler |
|---|---|---|---|
| $\rho = 0.1$ | $90.60_{\pm 0.02}$ | $90.05_{\pm 0.08}$ | $\mathbf{91.73_{\pm 0.13}}$ |
| $\rho = 0.2$ | $90.56_{\pm 0.02}$ | $88.96_{\pm 0.17}$ | $\mathbf{92.35_{\pm 0.26}}$ |
| $\rho = 0.3$ | $90.05_{\pm 0.27}$ | $87.89_{\pm 0.22}$ | $\mathbf{92.13_{\pm 0.15}}$ |
| $\rho = 0.4$ | $89.76_{\pm 0.19}$ | $86.79_{\pm 0.17}$ | $\mathbf{92.06_{\pm 0.10}}$ |

*Table 16.* CIFAR-100 test accuracy (%, mean $\pm$ std over 3 seeds) for ResNet-32 trained with USAM under three learning-rate schedules with $wd = 0.0$. Best in bold.

| $wd = 0.0$ | Constant USAM | USAM with Cosine Annealing | USAM with Stochastic Polyak Scheduler |
|---|---|---|---|
| $\rho = 0.1$ | $89.11_{\pm 0.07}$ | $88.27_{\pm 0.16}$ | $\mathbf{89.47_{\pm 0.15}}$ |
| $\rho = 0.2$ | $88.91_{\pm 0.20}$ | $86.81_{\pm 0.14}$ | $\mathbf{89.81_{\pm 0.23}}$ |
| $\rho = 0.3$ | $88.65_{\pm 0.06}$ | $85.76_{\pm 0.18}$ | $\mathbf{90.24_{\pm 0.13}}$ |
| $\rho = 0.4$ | $88.02_{\pm 0.16}$ | $84.49_{\pm 0.32}$ | $\mathbf{90.51_{\pm 0.11}}$ |

*Table 17.* CIFAR-10 test accuracy (%, mean $\pm$ std over 3 seeds) for ResNet-32 trained with SAM under three learning-rate schedules with $wd = 0.0$. Best in bold.

| $wd = 0.0$ | Constant SAM | SAM with Cosine Annealing | SAM with Stochastic Polyak Scheduler |
|---|---|---|---|
| $\rho = 0.1$ | $89.28_{\pm 0.06}$ | $88.28_{\pm 0.22}$ | $\mathbf{89.85_{\pm 0.09}}$ |
| $\rho = 0.2$ | $88.92_{\pm 0.01}$ | $86.75_{\pm 0.14}$ | $\mathbf{90.23_{\pm 0.13}}$ |
| $\rho = 0.3$ | $87.79_{\pm 0.38}$ | $84.73_{\pm 0.19}$ | $\mathbf{90.61_{\pm 0.15}}$ |
| $\rho = 0.4$ | $86.34_{\pm 0.12}$ | $80.61_{\pm 0.78}$ | $\mathbf{91.00_{\pm 0.12}}$ |

*Table 18.* CIFAR-10 test accuracy (%, mean $\pm$ std over 3 seeds) for ResNet-32 trained with SAM under three learning-rate schedules with $wd = 5 \cdot 10^{-4}$. Best in bold.

| $wd = 5 \cdot 10^{-4}$ | Constant SAM | SAM with Cosine Annealing | SAM with Stochastic Polyak Scheduler |
|---|---|---|---|
| $\rho = 0.1$ | $90.56_{\pm 0.20}$ | $90.27_{\pm 0.35}$ | $\mathbf{91.51_{\pm 0.28}}$ |
| $\rho = 0.2$ | $90.40_{\pm 0.21}$ | $88.99_{\pm 0.18}$ | $\mathbf{92.12_{\pm 0.12}}$ |
| $\rho = 0.3$ | $89.58_{\pm 0.07}$ | $86.75_{\pm 0.17}$ | $\mathbf{91.79_{\pm 0.22}}$ |
| $\rho = 0.4$ | $88.26_{\pm 0.38}$ | $84.21_{\pm 0.22}$ | $\mathbf{90.87_{\pm 0.12}}$ |

*Table 19.* CIFAR-100 test accuracy (%, mean $\pm$ std over 3 seeds) for ResNet-32 trained with SAM under three learning-rate schedules with $wd = 0.0$. Best in bold.

| $wd = 0.0$ | Constant SAM | SAM with Cosine Annealing | SAM with Stochastic Polyak Scheduler |
|---|---|---|---|
| $\rho = 0.1$ | $88.78_{\pm 0.22}$ | $88.30_{\pm 0.12}$ | $\mathbf{89.80_{\pm 0.13}}$ |
| $\rho = 0.2$ | $89.10_{\pm 0.04}$ | $86.93_{\pm 0.19}$ | $\mathbf{90.30_{\pm 0.16}}$ |
| $\rho = 0.3$ | $88.19_{\pm 0.14}$ | $84.53_{\pm 0.29}$ | $\mathbf{90.54_{\pm 0.09}}$ |
| $\rho = 0.4$ | $86.50_{\pm 0.06}$ | $82.31_{\pm 0.23}$ | $\mathbf{90.83_{\pm 0.15}}$ |

