# OpenReview forum: "Adaptive Sharpness-Aware Minimization with a Polyak-type Step size: A Theory-Grounded Scheduler"
_ICML.cc/2026/Conference — ICML 2026 regular_

### Official Review · Reviewer_tDgo · 2026-03-09

**Soundness:** 2
**Presentation:** 2
**Significance:** 2
**Originality:** 2
**Overall Recommendation:** 4
**Confidence:** 4

**Summary:**

This study introduces a Polyak-based stepsize for the sharpness-aware minimization (SAM) framework, referred to as the Polyak Scheduler. The authors provide a theoretical analysis of the convergence behavior of unnormalized SAM equipped with the Polyak Scheduler in both strongly convex and convex settings. They prove that, in deterministic scenarios, the method converges to the optimal solution, while in stochastic settings, it converges to a neighborhood around the optimum. Experimental results show that the proposed approach outperforms the conventional cosine annealing schedule.

**Compliance With Llm Reviewing Policy:**

Affirmed.

**Key Questions For Authors:**

see the weakness section.

**Limitations:**

yes

**Strengths And Weaknesses:**

### Strengths
- This paper applies Polyak stepsize to the Sharpness-aware minimization problem, proposing the Polyak Scheduler.
- This paper analyzes the convergence rate of Unnormalized Sharpness-aware minimization with Polyak scheduler in the (strongly-)convex setting. Then, they show that in the deterministic case, it converges to the optimal solution, and in the stochastic case, it converges to the neighborhood of the optimal solution.
- In the experiments, this paper demonstrated that the proposed method is better than cosine annealing.

### Weaknesses
- The proposed method does not converge to the optimal solution in the stochastic setting. There seem to be several prior works that proposed the Polyak-based methods, which can converge even in this stochastic setting [1,2]. Why were these methods not used? At the very least, it would be advisable to add discussions in this paper.
- The experiments are minimal. This paper evaluated the proposed method only with CIFAR-10/100 and ResNet-20. It would be better to evaluate the proposed method on at least three datasets and models, e.g., ImageNet, VGG, and a transformer-based model.
- This paper evaluated SGD with Polyak scheduler, showing that it can outperform SGD with cosine annealing. However, these days, Adam(W) and SGD with momentum are more commonly used than SGD. The reviewer is wondering if the idea of Polyak Scheduler is also applicable for other optimizers, such as Adam.

#### Reference
[1] Orvieto et. al., Dynamics of SGD with Stochastic Polyak Stepsizes: Truly Adaptive Variants and Convergence to Exact Solution, In NeurIPS 2022

[2] Jiang et. al., Adaptive SGD with Polyak stepsize and Line-search: Robust Convergence and Variance Reduction, In NeurIPS 2023

---

> ### Author Rebuttal · Authors · 2026-03-31
>
> **Thank you for the helpful references and suggestions.**
>
> **On the prior Polyak-based stochastic methods that converge exactly:**
> We thank the reviewer for the suggestion. Let us clarify below why simply using other variants of SPS (Orvieto et al and Jiang et al) is not necessarily beneficial for SAM-type methods. We are well aware of these works, and we can easily include a detailed discussion of them, along with the challenges of incorporating them into SAM-type methods, in our paper.
> Let us restate that the focus of our work differs from that of the cited papers. The methods of (Orvieto et al and Jiang et al) are designed for SGD-type updates, whereas in sharpness-aware optimization (SAM), the extrapolation point $e^t$​ changes both the update direction and the numerator of the Polyak scheduler, so the resulting step size and proof are not a direct reuse of the classical SPS variants. Having a decreasing SPS-type step size \gamma_k does not necessarily imply that the proposed Polyak SAM method will achieve exact convergence to the optimal solution, similar to SGD (Orvieto et al. and Jiang et al.). In this work, our goal was to establish the *first* convergence guarantees for Polyak-type step sizes in the SAM framework.
> Having said that, we agree with the reviewer that designing a sharpness-aware adaptive method with exact convergence guarantees would be an interesting future direction.
>
>
> **On the empirical scope**, we agree that a larger-scale evaluation would strengthen the paper. At the same time, the current experimental section is broader than your summary suggests: beyond the synthetic experiments (ridge regression), the paper reports CIFAR-10/100 results with both ResNet-20 and ResNet-32, with additional tables in Appendix E. Evaluation on larger datasets/models such as ImageNet or transformer-based architectures is in our to-do list, but it is not feasible to complete it during the rebuttal window. We commit to include at least one more larger setting in the camera-ready version of our work.
>
> **Regarding other optimizers such as Adam or momentum variants:** Thanks for the suggestion, indeed, combining Polyak and Adam is a valid and interesting future direction.
> For momentum, the recent paper Oikonomou & Loizou (2025c)  has already combined Polyak-type schedulers with SGD with momentum.
> In our work, we focus mostly on SAM-type methods, and in our numerical evaluation, we wanted to show that SAM with the Polyak scheduler is better than the popular cosine annealing in the SAM literature. Our experiments successfully served this goal. That said, extending Polyak-style sharpness-aware schedulers to momentum or Adam-based variants is a promising direction, and we will mention this more explicitly.
>
> **Thanks again for the review and the positive evaluation of our work.
> After reading your comments, we believe all the pointed weaknesses are simple clarifications. If you agree that we managed to address all issues, please consider raising your mark to clearly support our work. If you believe this is not the case, please let us know so that we have a chance to respond.**

---

> > ### Author Rebuttal · Reviewer_tDgo · 2026-04-02
> >
> > The authors promised to add additional experiments with a larger dataset. I also recommend the authors to add the discussion about [1,2]. Overall, my main concers have been addressed by the rebuttal. I would like to keep my original score

---

> > > ### Author Response · Authors · 2026-04-02
> > >
> > > **We thank the reviewer for engaging with our rebuttal and for confirming that the main concerns have been addressed.** We will include experiments on a larger dataset as well as a detailed discussion of [1,2] in the camera-ready version of our paper.

---

### Official Review · Reviewer_mfHi · 2026-03-09

**Soundness:** 3
**Presentation:** 3
**Significance:** 3
**Originality:** 3
**Overall Recommendation:** 4
**Confidence:** 3

**Summary:**

This paper studies Polyak-type learning-rate schedulers for sharpness-aware optimization, with the main focus on USAM. The proposed schedulers adapt the classical Polyak/SPSmax idea to updates based on gradients evaluated at perturbed points. The paper develops both deterministic and stochastic variants, shows that they reduce to the classical Polyak rules when $\rho=0$, proves linear convergence in squared distance under strong convexity, and gives $O(1/T)$-type averaged gradient-norm guarantees in convex smooth settings, up to a neighborhood in the stochastic case. The paper also discusses interpolation and constant-step corollaries, and presents synthetic and CIFAR experiments to support their findings.

**Compliance With Llm Reviewing Policy:**

Affirmed.

**Final Justification:**

I thank the authors for the rebuttal. My overall assessment remains unchanged, and I continue to view the paper positively.

**Key Questions For Authors:**

- It would be helpful to understand how large the gap is between the algorithm used in the experiments and the one analyzed theoretically. Could the authors provide an ablation or comparison to quantify this difference?
- It would also be helpful to clarify whether the method can be made universal, in the sense of not requiring knowledge of the smoothness constant L.

**Limitations:**

Yes.

**Strengths And Weaknesses:**

Strengths:

- The main idea is well motivated. Using Polyak-style, function-value-based adaptation for sharpness-aware updates seems like a natural direction, and, to the best of my knowledge, a plausibly novel one. The reduction to the classical Polyak/SPSmax rule at $\rho=0$ is also conceptually clean and helps place the method within the broader adaptive-stepsize literature.
- The deterministic and stochastic analyses are generally easy to follow. Under the stated assumptions, the descent-style arguments are fairly transparent, and the strongly convex and convex guarantees are presented clearly.
- The appendix includes substantial additional experimental details to support their findings.

Weaknesses:

-  The experiments do not correspond to the analyzed method. The stochastic USAM method is stated as $x_{t+1} = x_t - \gamma_t \nabla f_{S_t}(e_t).$ However, Algorithm 1 in the appendix instead updates with $\nabla f_{S_t}(x_t)$ rather than $\nabla f_{S_t}(e_t)$, and its numerator introduces an additional factor of $\rho_t$ relative to the stated formula.

---

> ### Author Rebuttal · Authors · 2026-03-31
>
> **We thank the reviewer for their comments.**
>
> **On the implementation of the algorithm:** Thanks for identifying the typo in the appendix. We will make sure it is fixed in the camera-ready version. To clarify, the method analyzed throughout the paper is the stochastic USAM update described in lines 92-93, combined with the stochastic Polyak scheduler described in lines 234 and 259. This is the same method implemented in the experiments, as can be verified from the code provided in the supplementary zip.
>
> As such, the point cited as a weakness in our work is simply a typo that can be easily corrected in the camera-ready version. To also clearly answer the reviewers' first question, there is no gap between the algorithm used in the experiments and the one analyzed theoretically. They are both the same method.
>
> **On universality:** Yes, this is an interesting direction, and in Section 3, we give a partial answer to the question of universality (not requiring knowledge of smoothness constant L). In particular, Theorem 3.4 shows that in the deterministic convex setting, one can avoid fixing a radius via a known $L$ by using any decreasing sequence $\rho_t\downarrow0$, while still guaranteeing convergence. Extending this idea to the stochastic setting, and potentially to non-convex problems, is a natural next step. We consider this a promising direction for future work.
>
> **Thanks again for the review and the positive evaluation of our work.
> After reading your comments, we believe the identified weaknesses are simple clarifications (a typo in the algorithm description in the appendix). If you agree that we managed to address all issues, please consider raising your mark to clearly support our work. If you believe this is not the case, please let us know so that we have a chance to respond.**

---

> > ### Author Rebuttal · Reviewer_mfHi · 2026-04-02
> >
> > I thank the authors for their responses and would like to maintain my positive assessment.

---

> > > ### Author Response · Authors · 2026-04-02
> > >
> > > **We thank the reviewer for the positive assessment and for the constructive feedback.**

---

### Official Review · Reviewer_3Rse · 2026-03-11

**Soundness:** 3
**Presentation:** 3
**Significance:** 2
**Originality:** 3
**Overall Recommendation:** 3
**Confidence:** 4

**Summary:**

**Summary**

This paper applies (stochastic) Polyak stepsize to tune parameters in sharpness-aware minimization. The authors show convergence guarantees for both deterministic and stochastic convex functions. Some experiments showcase the effectiveness of the results.

**Compliance With Llm Reviewing Policy:**

Affirmed.

**Final Justification:**

In the rebuttal, the authors promised to discuss the concerns as limitations. However, these concerns remain largely unaddressed, except for some explanations of the technical difficulties. I therefore maintain my evaluation of the paper. But I won't object if the rest of the review team leans towards it.

**Key Questions For Authors:**

**Minor issues**

1. Line 382

   Typo in the parameter.

**Questions**

1. Could you show how increasing $\rho_k$ benefits generalization theoretically?

**Limitations:**

Yes.

**Strengths And Weaknesses:**

**Strengths**

The paper is quite well-written. The idea is to leverage the intermediate point from USAM in the distance upperbound and use the principle of distance minimization to derive a closed-form stepsize.

**Weaknesses**

There are two major weaknesses.

1. Setting mismatch

   Since the Polyak stepsize argument relies on convexity to replace $x^\star$ in the inner product by $f(x^\star)$, this paper only considers analysis in the convex case. However, I don't think convex problems are where sharpness-aware minimization is typically applied to.

2. Worse convergence rates and remaining parameter dependence

   The convergence rate developed in the paper does not justify the practical performance of the approach. In particular, the condition number dependence is squared in the strongly convex case, while for the general convex case, the guarantee is stated only with respect to the gradient norm instead of suboptimality $f(x) - f(x^\star)$. These are the costs of having a closed-form stepsize formula, but they are worth discussing. Finally, the dependence on $\rho_t$ persists and requires tuning, and it's not satisfying.

Overall, the paper's analysis is standard in SPS literature. The experiments look promising, while the theoretical guarantees are not. I currently recommend weak rejection, and I'm willing to raise my score if the authors address my concerns (rate improvement/setting generalization)

---

> ### Author Rebuttal · Authors · 2026-03-31
>
> **We thank the reviewer for their comments.**
>
> **On convexity:**
> We agree with the reviewer's comment on the use of convexity in our convergence guarantees. We will be happy to provide more details on this theory-practice gap in the camera-ready version of our paper. The reason for focusing on the (strongly) convex regime is that the Polyak-style derivation used here fundamentally relies on convex structure to replace the unknown optimizer $x^*$ by function values, and the proofs then build on this same mechanism.  In addition, convexity is used in establishing the descent property in our Proposition 2.2. We would be happy to explicitly mention this in our updated version and also include the specific parts (specific technical hurdles) of the proofs that prevent us from providing convergence guarantees for non-convex objectives.
> The use of convexity in our proofs should be seen as a feature that allows us to provide novel step-size selection based on Polyak-type ideas, rather than as a limitation. Having said that, we should also note that convexity is common in the SAM literature, even if the algorithm is originally designed for non-convex problems: see, for example, the recent works of Andriushchenko (2022), Dai (2023), Si (2023), and Khanh (2024).
>
> **On the convergence rates:**
> Yes, we agree with the reviewer  that our proposed convergence rates do not justify the practical performance of the approach. The slower convergence, as the reviewer correctly pointed out, is an artifact of the adaptive step-size selection (\gamma_k), which itself does not depend on parameters L or \mu. We would be happy to discuss the details and limitations of our convergence guarantees more thoroughly in the camera-ready version of our work.
> Our setting combines two sources of difficulty compared to classical SGD-type methods in the convex regime: (i) the extrapolation step $e^t$, and the (ii) Polyak adaptivity. We believe that some trade-off in the resulting rates is natural, considering these challenges. Moreover, using the gradient norm guarantees in the convex setting (instead of $f(x^t)-f(x^*)$) is not unusual in the SAM literature: for example, [Xie, 2024] establishes guarantees in the smooth convex setting in terms of the gradient norm, rather than the suboptimality gap. As we already mentioned, we can easily add such a discussion in the camera-ready version. We therefore believe that, despite the conservative nature of the current guarantees, the ideas behind our adaptive step-size are novel for the SAM methods, and its strong practical performance makes it a worthwhile contribution to share with the ML community.
>
> **On choice of $\rho$:** Theorem 3.4 already provides a setting in which $\rho_t$ is decreasing, and in that case the choice of$\rho_t$ does not require knowledge of L. Regarding the increasing values of $\rho$ in the experimental tables, we observe that, for both standard SAM with a constant learning rate and SAM with cosine annealing, generalization performance tends to decrease as $\rho$ increases. The same trend is present for SAM with the Polyak scheduler, although in that case the best performance is achieved at $\rho=0.2$ rather than $\rho=0.1$, which was the best choice for the other schedulers.
>
> **If you agree that we managed to address all issues, please consider raising your mark to support our work. From our viewpoint, all concerns raised are mainly clarifications that can be addressed in the camera version of our work, and not reasons justifying a rejection score.  If you believe this is not the case, please let us know so that we have a chance to respond.**

---

> > ### Author Rebuttal · Reviewer_3Rse · 2026-04-01
> >
> > Thank you for the response. Although the authors promise to discuss these concerns as limitations, they remain essentially unaddressed. Overall, I still think this is a well-written paper, but these limitations should either be addressed or be accompanied by a hardness result. I maintain my evaluation of the paper, but won't strongly object if the rest of the review team decides to accept it.

---

> > > ### Author Response · Authors · 2026-04-02
> > >
> > > **We thank the reviewer for engaging with our rebuttal and for the kind words about the writing quality of our paper.**
> > >
> > > We would like to respectfully ask for clarification on which specific concerns the reviewer considers "essentially unaddressed", so that we can respond as precisely as possible during the remaining discussion period. In our opinion, the concerns raised by the reviewer were mostly clarifications that we addressed in our original response.
> > >
> > > To our understanding, the reviewer raised two main concerns:
> > >
> > > **1. Setting mismatch (convexity).** As we mentioned in our original response, convexity is fundamental to the Polyak-style derivation, which replaces the unknown $x^\ast$ by the function value $f(x^\ast)$. We also noted that convexity is standard across the SAM convergence literature (Andriushchenko 2022, Dai 2023, Si 2023, Khanh 2024).
> > > In our opinion, even if SAM was originally designed for DNNs with non-convex losses, using convexity to derive a new update rule that also works for DNNs in practical scenarios (as we show in our experiments) should be sufficient to justify novelty and be of interest to the general ML community. In the end, many recent papers show that convex optimization algorithms and theory describe the training of modern DNNs and LLMs well. See, for example, [1].
> > >
> > > Let us also provide a discussion explaining why our current approach cannot trivially be extended to the non-convex regime (in our opinion this is not a drawback of our paper):
> > > The techniques used in the convex setting, i.e., where we expanded $\|x^{t+1}-x^\ast\|^2-\|x^t-x^\ast\|^2$, do not work anymore because there is no way to lower bound the inner product $\langle\nabla f(e^t),e^t-x^\ast\rangle$ without convexity. The standard alternative in the non-convex case is to use the smoothness descent (equation (10) in our document) for $(x,y)=(x^{t+1},x^t)$ to get the following inequality
> > > $f(x^{t+1})\leq f(x^t)-\gamma_t\langle\nabla f(e^t),\nabla f(x^t)\rangle+\frac{L\gamma_t^2}{2}\|\nabla f(e^t)\|^2$. One can bound the quantities $\langle\nabla f(e^t),\nabla f(x^t)\rangle$ and $\|\nabla f(e^t)\|^2$ by $\|\nabla f(x^t)\|$ using only smoothness (see Lemma B.7) but the numerator of the Polyak step size introduces the term $f(e^t)-f^\ast$ into the descent bound, and relating this back to $\|\nabla f(x^t)\|^2$ requires again convexity-type arguments.
> > >
> > > We would be happy to include a discussion on non-convex problems in the updated version of our work.
> > >
> > > Could you please clarify what concrete form of "addressing" this concern would take beyond the points we shared above?
> > >
> > >
> > > **2. On the convergence rates.** We appreciate the reviewer's feedback, and as we originally mentioned in our response, we plan to include a discussion in the camera-ready version of our work that highlights the benefits and limitations of our results and the reasoning behind the slower rates. In particular, we plan to include the following:
> > >
> > > Limitations of Theorem 3.1:
> > >
> > > “In the strongly convex case, the contraction factor $1 - \mu^2(1-L\rho)^2/(4L^2)$ has a squared dependence on the condition number $\kappa = L/\mu$, in contrast to the $1 - O(1/\kappa)$ rate typically achieved by gradient descent or SAM with constant step size. This gap arises from the following: The descent property (Proposition 2.2) bounds the quantity $\|x^{t+1}-x^\ast\|^2-\|x^t-x^\ast\|^2$ in terms of $\|\nabla f(x^t)\|^2$ rather than $\|x^t - x^\ast\|^2$ directly. This introduces the square on $\kappa$ when converting via the error bound $\|\nabla f(x^t)\|^2 \geq \mu^2\|x^t - x^\ast\|^2$.”
> > >
> > > Limitations of Theorem 3.2:
> > >
> > > “In the convex case, the guarantee is stated in terms of the averaged squared gradient norm $\frac{1}{T}\sum_{t=0}^{T-1}\|\nabla f(x^t)\|^2$ rather than the suboptimality gap $f(x^T) - f^\ast$. This is a consequence of the same descent property: Proposition 2.2  bounds the quantity $\|x^{t+1}-x^\ast\|^2-\|x^t-x^\ast\|^2$ in terms of $\|\nabla f(x^t)\|^2$, and telescoping this bound naturally yields a guarantee on the averaged gradient norms. Obtaining a suboptimality gap bound would require relating $f(x^t) - f^\ast$ to $\|\nabla f(x^t)\|^2$ from above, which in the convex (non-strongly convex) case is not possible without additional assumptions.”
> > >
> > > **With the above two points handled, we believe our paper would clearly explain the limitations of our results. We thank the reviewer for these suggestions.**
> > >
> > > **From our viewpoint, all concerns raised by the reviewer are primarily clarifications that can be addressed in the camera-ready version of our work (we explain how with our response), rather than reasons justifying a rejection score.**
> > >
> > > [1] Schaipp, Fabian, Alexander Hägele, Adrien Taylor, Umut Simsekli, and Francis Bach. "The Surprising Agreement Between Convex Optimization Theory and Learning-Rate Scheduling for Large Model Training." In ICML 2025.

---

### Official Review · Reviewer_DUhi · 2026-03-12

**Soundness:** 3
**Presentation:** 3
**Significance:** 2
**Originality:** 3
**Overall Recommendation:** 4
**Confidence:** 4

**Summary:**

The submission attempts to consider a central concept by adapting the classical Polyak step size and its stochastic counterpart (SPS) to the sharpness-aware optimization framework, thereby minimizing the burden of hyperparameter tuning. By upper-bounding the distance to the optimum, the authors derive a closed-form, adaptive step size that depends only on the current mini-batch loss and the gradient at the perturbed point. The paper provides rigorous convergence guarantees for convex and strongly convex objectives in both deterministic and stochastic settings. Finally, the authors empirically validate their proposed scheduler on synthetic ridge regression and image classification tasks (CIFAR-10/100 with ResNets), demonstrating that it matches or outperforms carefully tuned constant and cosine-annealing baselines, particularly exhibiting robustness to larger sharpness radii.

**Compliance With Llm Reviewing Policy:**

Affirmed.

**Key Questions For Authors:**

See the weaknesses above

**Limitations:**

Yes

**Strengths And Weaknesses:**

# Strengths

- Novel Connection: Bridging Polyak step sizes with SAM-type updates is unexplored. The derivation of the step size is mathematically clean and intuitively extends the classical Polyak rule to the perturbed geometry of SAM.
- Strong Theoretical Foundation: The authors manage to prove linear convergence for strongly convex objectives and  O(1/T) rates for convex objectives without relying on restrictive assumptions commonly found in adaptive SAM literature, such as bounded gradients, bounded variance, or complex growth conditions. The introduction of the variance measure 𝜎 quantify the convergence neighborhood is standard and well-handled.
- Empirical Robustness: The experiments highlight an interesting property of the proposed Stochastic Polyak Scheduler: it is  more robust to the choice of the sharpness radius 𝜌 compared to standard cosine annealing.

# Weaknesses

While the paper is well-written and the core idea is clear, there are a few areas in both theory and experiments that need further clarification and strengthening.

- The non-convexity gap: The theoretical guarantees are strictly confined to convex and strongly convex functions. However, the primary motivation and empirical application of SAM is the training of highly non-convex Deep Neural Networks (DNNs). While this theory-practice gap is common in optimization literature, the authors should explicitly acknowledge it. It would significantly strengthen the paper if the authors could provide at least a stationary point convergence guarantee (e.g., bounding the gradient norm) for non-convex objectives, or discuss the specific technical hurdles preventing this.
- Smoothness assumption (𝜌≤1/𝐿): Proposition 2.1 establishes the non-negativity of the step size (allowing the removal of the ReLU safeguard) under the condition  ρ≤1/L. In deep learning, the local Lipschitz constant L can be extremely large and fluctuates wildly during training. It is unlikely that a constant ρ∈{0.1,0.2,0.3,0.4} satisfies this globally.
- Question for authors: How often does the ReLU safeguard [⋅] + actually activate during the ResNet training on CIFAR? Adding an empirical plot tracking the activation frequency of the safeguard or estimating the local L would provide great insight into the algorithm's practical dynamics.
- Experimental Baselines: The empirical evaluation compares the proposed method against constant learning rates and cosine annealing. However, the introduction explicitly mentions several recent adaptive SAM methods (e.g., AdaSAM, SAM with Adagrad/Adam, or Naganuma et al., 2024). To truly demonstrate the superiority of the Polyak approach, the authors should include at least one state-of-the-art adaptive SAM baseline in their CIFAR experiments.
- Sensitivity to the hyperparameter (γb):The paper claims to reduce the learning-rate tuning burden. However, the stochastic scheduler introduces a new capping hyperparameter γb. According to Table 5 in the appendix, γ​ was tuned over a massive grid (10*-3 to 10*3 ).
Question for authors: Does the performance heavily depend on the exact choice of 𝛾𝑏? An ablation study showing the sensitivity of the final test accuracy to 𝛾𝑏 is necessary to substantiate the claim that this method is easier to tune than standard schedulers.
- Assumption on ℓSt∗ =0: In the DNN experiments, the authors set the lower bound is 0. This works well for standard cross-entropy loss on clean data where the model can interpolate. However, how does the scheduler behave in non-interpolated regimes (e.g., datasets with high label noise)? A brief discussion or a small experiment on CIFAR with label noise would make the empirical section much more comprehensive.

# Minor Comments and Typos

- Line 279: "Contary to the previous theorem..." should be "Contrary to the previous theorem...".
- Table 1 is very helpful and well-organized, but please ensure that the definition of 𝑁 (neighborhood) is clearly distinguished between the different cited works in the main text, as they rely on different variance assumptions.
- In Algorithm 1 (Appendix E.2), step 4 includes the ReLU safeguard implicitly via the max operation if 𝛾𝑏 is the only bound, but the text states [⋅]+ is used for SAM. Please ensure the pseudocode perfectly matches the implemented equations for both USAM and SAM.

---

> ### Author Rebuttal · Authors · 2026-03-31
>
> **Thank you for the positive assessment of the novelty, theory, and robustness results.**
>
> **On the non-convexity gap:**
> We agree with the reviewer's comment on the use of convexity to our convergence guarantees. We will be happy to provide more details on this theory-practice gap in the camera-ready version of our paper.
> The reason for focusing on the (strongly) convex regime is that the Polyak-style derivation used here fundamentally relies on convex structure to replace the unknown optimizer $x^\ast$ by function values, and the proofs then build on this same mechanism.  In addition, convexity is used in establishing the descent property in our Proposition 2.2. We will explicitly mention this in our updated version and also include the specific parts (specific technical hurdles) of the proofs that prevent us from providing a stationary-point convergence guarantee for non-convex objectives.
> We agree that this assumption is a limitation of the current theory. However, we should point out that convexity is common in the SAM literature: see, for example, the recent works of Andriushchenko (2022), Dai (2023), Si (2023), and Khanh (2024).
>
> **On smoothness**: We agree that the condition $\rho\leq1/L$ is not realistic in the deep learning setting. This condition is used to establish nonnegativity of the numerator and, consequently, to identify when the ReLU safeguard can be removed analytically. Our DNN experiments, however, provide empirical evidence that the resulting scheduler remains useful beyond the regime covered by the theory. Let us note that the smoothness assumption is standard in the theoretical SAM literature. For example, all closely related works listed in Table 1 rely on smoothness assumptions (which typically appear as a restriction on $\rho$ and $\gamma$ and in the main theorems).
>
> **On ReLU safeguard:** Thank you for the suggestion. As already noted (lines 422-423), the safeguard is rarely active in our DNN experiments, and we agree that an explicit step size plot would make this much clearer. We will add this to the camera-ready version (and we can include such plots for all settings). As an initial observation, we note that for ResNet-32 on CIFAR-100, the ReLU safeguard is **never** activated.
>
> **On adaptive-SAM baselines**, this is a fair request. Our main empirical goal was to compare the proposed Polyak scheduler against standard schedule choices already used with SAM/USAM, namely, tuned constant steps and cosine annealing. We would like to compare with more recent papers on adaptive SAM, like AdaSAM, SAM with Adagrad/Adam, and Naganuma et al. (2024), as we agree that this will be beneficial for our work. However, at the time of finalizing our work, none of these papers had their implementation available online (e.g., GitHub repo). We needed to recreate their algorithms from scratch, and this would have led to potentially misleading information (compared to what is reported in this work). We commit to reaching out to the authors of these papers to request their implementation. If provided, we would be happy to have a detailed comparison between all existing adaptive SAM methods. We agree with the reviewers that this will strengthen the numerical evaluation of our work.
>
> **On sensitivity of $\gamma_b$:** We performed a broad tuning sweep for several of our settings. For example, on ResNet-20 on CIFAR-10, the best test accuracy obtained on ResNet-20/CIFAR-10 for each value of $\gamma_b$ is as follows:
> - $\gamma_b=10^{-3}$: 53.33 %
> - $\gamma_b=10^{-2}$: 79.39 %
> - $\gamma_b=10^{-1}$: 88.38 %
> - $\gamma_b=10^{0}$: 88.87 %
> - $\gamma_b=10^{1}$: 87.86 %
> - $\gamma_b=10^{2}$: 87.49 %
> - $\gamma_b=10^{3}$: 87.48 %
>
> These results suggest that performance is fairly stable once $\gamma_b$ is chosen in a reasonable range, with $\gamma_b=1$ giving the best overall result in our sweep. Based on these observations, we selected $\gamma_b=1.0$, and then used this same value for all remaining experiments. This is also the value we would recommend in practice.
>
> **On $\ell^\ast_{S_t}=0$:** For the DNN experiments, setting $\ell^\ast_{S_t}=0$ is a natural choice because cross-entropy is non-negative. Recall that $\ell^\ast_{S_t}$ is just a lower bound of the loss. In practise, since (almost) all commonly used losses are nonnegative, choosing $\ell^\ast_{S_t}=0$ is both valid and effective. We should highlight that this is not an assumption but a property of our problem (lower bounded by zero)
>
> Finally, thank you for the suggestions about Table 1 and the pseudocode. We will make sure they are fixed in the camera-ready version.
>
> **Thanks again for the review and the positive evaluation of our work.
> After reading your comments, we believe all the pointed weaknesses are simple clarifications. If you agree that we managed to address all issues, please consider raising your mark to clearly support our work. If you believe this is not the case, please let us know so that we have a chance to respond.**

---

> > ### Author Rebuttal · Reviewer_DUhi · 2026-04-02
> >
> > Thank you for the response. Although the authors promise to discuss these issues as limitations, my primary concerns regarding the convexity assumption, the lack of adaptive-SAM baselines, and the lower bound remain essentially unaddressed.

---

> > > ### Author Response · Authors · 2026-04-02
> > >
> > > **We thank the reviewer for the continued engagement and for acknowledging the quality of our work.** We would like to respectfully ask for clarification on which concerns the reviewer considers "essentially unaddressed," as we believe our rebuttal provided responses to each point raised in the original review. Let us briefly reiterate:
> > >
> > > **Convexity assumption.** As discussed in our previous response, convexity is fundamental to the Polyak-style derivation, which replaces the unknown $x^\ast$ by the function value $f(x^\ast)$. We also noted that convexity is standard across the SAM convergence literature (Andriushchenko 2022, Dai 2023, Si 2023, Khanh 2024). In our opinion, even if SAM was originally designed for DNNs with non-convex losses, using convexity to derive a new update rule that also works for DNNs in practical scenarios (as we show in our experiments) should be sufficient to justify novelty and be of interest to the general ML community. In the end, many recent papers show that convex optimization algorithms and theory describe the training of modern DNNs and LLMs well. See, for example, [1].
> > >
> > > Let us also provide a discussion explaining why our current approach cannot trivially be extended to the non-convex regime (in our opinion this is not a drawback of our paper):
> > > The techniques used in the convex setting, i.e., where we expanded $\|x^{t+1}-x^\ast\|^2-\|x^t-x^\ast\|^2$, do not work anymore because there is no way to lower bound the inner product $\langle\nabla f(e^t),e^t-x^\ast\rangle$ without convexity. The standard alternative in the non-convex case is to use the smoothness descent (equation (10) in our document) for $(x,y)=(x^{t+1},x^t)$ to get the following inequality
> > > $f(x^{t+1})\leq f(x^t)-\gamma_t\langle\nabla f(e^t),\nabla f(x^t)\rangle+\frac{L\gamma_t^2}{2}\|\nabla f(e^t)\|^2$. One can bound the quantities $\langle\nabla f(e^t),\nabla f(x^t)\rangle$ and $\|\nabla f(e^t)\|^2$ by $\|\nabla f(x^t)\|$ using only smoothness (see Lemma B.7) but the numerator of the Polyak step size introduces the term $f(e^t)-f^\ast$ into the descent bound, and relating this back to $\|\nabla f(x^t)\|^2$ requires again convexity-type arguments.
> > >
> > > We would be happy to include a discussion on non-convex problems in the updated version of our work.
> > >
> > > Could you please clarify what concrete form of "addressing" this concern would take beyond the points we shared above?
> > >
> > > **Adaptive-SAM baselines.** As we mentioned in our original response, none of the recently proposed adaptive SAM methods (AdaSAM, SAM with Adagrad/Adam, Naganuma et al. 2024) had publicly available implementations, and reimplementing them from scratch risked producing misleading comparisons.  We committed to reaching out to the respective authors and including these comparisons in the camera-ready version. In our opinion, the lack of an official and efficient codebase for prior work on the topic indicates that, even if the proposed algorithms are adaptive in nature, their implementation might require substantial tuning to make them practical. In our case, we include code with our work, showing exactly how our method can be implemented and be reproducible and we include experiments showing also the benefits of our approach in both convex and non-convex regimes.
> > >
> > > Based on the above, what exactly did the reviewer expect that sufficient experiments should be? Should we have recreated all previous algorithms from scratch and tuned them to their most efficient versions for each problem we tried?
> > > As we mentioned, we would be happy to do so, but we believe this would be reasonable if the code for prior work is available (otherwise, the comparison could be unfair).
> > >
> > > **Lower bound ($\ell^\ast_{S_t} = 0$)** We already clarified that setting $\ell^\ast_{S_t}=0$ is not an assumption but a condition always satisfied for all settings of interest. This should not be seen as a limitation of our results (this is not something that required tuning). For example, cross-entropy (and virtually all commonly used losses) are non-negative. The quantity $\ell^\ast_{S_t}$ is simply *any* lower bound on the mini-batch loss, and zero is always valid for non-negative losses. Could the reviewer clarify what specific concern remains on this point?
> > >
> > > **Let us thank the reviewer for engaging in a discussion with us and for the positive evaluation. If you believe that with the above, we still have not addressed your concerns in a clear way, please let us know so that we have a chance to respond.**
> > >
> > > [1] Schaipp, Fabian, Alexander Hägele, Adrien Taylor, Umut Simsekli, and Francis Bach. "The Surprising Agreement Between Convex Optimization Theory and Learning-Rate Scheduling for Large Model Training." In ICML 2025.

---

### Decision · Program_Chairs · 2026-04-30

**Decision:**

Accept (regular)

**Comment:**

The reviewers raised concerns that  the convexity assumption, the lack of adaptive-SAM baselines, and the lower bound are not addressed in the paper. The hardness results would potentially strengthen it as well. Experimental evaluation should be expanded to make the results more convincing.